# Monte Carlo Tree Search based Space Transfer for Black-box Optimization

**Shukuan Wang**[1,2][*], **Ke Xue**[1,2][*], **Lei Song**[1,2], **Xiaobin Huang**[1,2], **Chao Qian**[1,2][†]

[1]National Key Laboratory for Novel Software Technology, Nanjing University
[2]School of Artificial Intelligence, Nanjing University
{wangsk, xuek, huangxb, songl, qianc}@lamda.nju.edu.cn

## Abstract

Bayesian optimization (BO) is a popular method for computationally expensive black-box optimization. However, traditional BO methods need to solve new problems from scratch, leading to slow convergence. Recent studies try to extend BO to a transfer learning setup to speed up the optimization, where search space transfer is one of the most promising approaches and has shown impressive performance on many tasks. However, existing search space transfer methods either lack an adaptive mechanism or are not flexible enough, making it difficult to efficiently identify promising search space during the optimization process. In this paper, we propose a search space transfer learning method based on Monte Carlo tree search (MCTS), called MCTS-transfer, to iteratively divide, select, and optimize in a learned subspace. MCTS-transfer can not only provide a well-performing search space for warm-start but also adaptively identify and leverage the information of similar source tasks to reconstruct the search space during the optimization process. Experiments on synthetic functions, real-world problems, Design-Bench and hyperparameter optimization show that MCTS-transfer can demonstrate superior performance compared to other search space transfer methods under different settings. Our code is available at `https://github.com/lamda-bbo/mcts-transfer`.

## 1 Introduction

In many real-world tasks such as neural architecture search [56, 41, 42], hyper-parameter optimization [55, 4], and integrated circuit design [19, 30], we often need to solve black-box optimization (BBO) problems, where the objective function has no analytical form and can only be evaluated by different inputs, regarded as a "black-box" function. BBO problems are often accompanied by expensive computational costs of the evaluations, requiring a BBO algorithm to find a good solution with a small number of objective function evaluations.

Bayesian optimization (BO) [29, 8] is a widely used sample-efficient method for such problems. At each iteration, BO fits a surrogate model, typically Gaussian process (GP) [26], to approximate the objective function, and maximizes an acquisition function to determine the next query point. Under the limited evaluation budget, traditional BO methods only have a few observations, which are, however, insufficient for constructing a precise surrogate model, leading to slow convergence. Thus, traditional BO methods struggle to effectively solve expensive BBO problems, preventing their broader applications.

To tackle this issue, recent research tries to apply transfer learning methods for BO [2]. Transfer BO methods operate under the assumption that similar tasks may share common characteristics, and

---

[*]Equal Contribution
[†]Corresponding Author

38th Conference on Neural Information Processing Systems (NeurIPS 2024).

the knowledge acquired from similar source tasks can be helpful for optimizing the current target task. They typically gather offline datasets from the source tasks and utilize them to expedite the target task. These methods can be categorized based on the learned components of BO, such as the acquisition function [51, 40], initialization point [45, 47], or search space [49, 23, 17].

Among these, learning the search space is a promising research area due to its effectiveness and orthogonal relationship with optimization methods. By learning and partitioning the search space, we can more effectively utilize potential subspaces and significantly accelerate the algorithm's search for optimal solutions. While the methods for partitioning the search space have demonstrated great potential [22, 14, 23, 17], they still have several limitations, particularly in terms of flexibly adjusting the search space for the current target task. In many scenarios, we are uncertain about which tasks are truly "similar" to the target task before optimizing it. Our comprehension about this similarity can only deepen gradually as we progress in optimizing the target task. Therefore, we hope that a search space transfer algorithm can automatically identify the most relevant source tasks during the optimization process, and give them more consideration when constructing the subspaces. However, existing methods have limited flexibility in adjusting in this manner.

In this paper, we propose a search space transfer learning method based on Monte Carlo tree search (MCTS), called MCTS-transfer, to iteratively divide, select, and optimize in a learned subspace. Each node of MCTS-transfer represents a subspace, whose potential is calculated as a weighted sum of the source and target sample values, assessing the node's utilization value. To better identify and leverage the information from source tasks, we assign different weights to different source tasks based on their similarity to the target task, which are adjusted dynamically during the optimization process. These weights are then used to calculate the potential of the node. Our proposed MCTS-transfer offers several notable advantages. First, it can provide a better initial search space for a warm-start of the optimization of the target task. Second, it can provide multiple promising compact subspaces by MCTS, to improve optimization efficiency. The upper confidence bound (UCB)-based node selection of MCTS can also automatically balance the trade-off between exploration and exploitation. Third, it can automatically identify source tasks similar to the current target task based on new observations, and re-construct the search space to make it more consistent with the current target task.

To evaluate the effectiveness of the proposed method, we compare MCTS-transfer with various search space transfer methods and conduct experiments on multiple BBO tasks, including synthetic functions, real-world problems, Design-Bench and hyper-parameter optimization problems. In different scenarios, such as varying similarity between the source tasks and target task, MCTS-transfer demonstrates superior performance. We also analyze the effectiveness of the adaptive weight adjustment, showing that MCTS-transfer can identify source tasks that are similar to the target task and assign them higher weights accordingly. Note that MCTS-transfer can be combined with any BBO algorithm. We only implemented it with the basic BO algorithm (i.e., using GP as the surrogate model and expected improvement (EI) as the acquisition function) in the experiments, and the performance can be further enhanced by advanced techniques.

## 2 Background

### 2.1 Bayesian Optimization

We consider the problem $\max_{\boldsymbol{x} \in \mathcal{X}} f(\boldsymbol{x})$, where $f$ is a black-box function and $\mathcal{X} \subseteq \mathbb{R}^D$ is the domain. The basic framework of BO contains two critical components: a surrogate model and an acquisition function. GP is the most popular surrogate model. Given the sampled data points $\{(\boldsymbol{x}_i, y_i)\}_{i=1}^{t-1}$, where $y_i = f(\boldsymbol{x}_i) + \epsilon_i$ and $\epsilon_i \sim \mathcal{N}(0, \eta^2)$ is the observation noise, GP at iteration $t$ seeks to infer $f \sim \mathcal{GP}(\mu(\cdot), k(\cdot, \cdot) + \eta^2 \mathbf{I})$, specified by the mean $\mu(\cdot)$ and covariance kernel $k(\cdot, \cdot)$, where $\mathbf{I}$ is the identity matrix of size $D$. After that, an acquisition function, e.g., probability of improvement (PI) [15], EI [12] or UCB [36], is optimized to determine the next query point $\boldsymbol{x}_t$, balancing exploration and exploitation.

### 2.2 Transfer Bayesian Optimization

When solving new BBO problems, traditional BO methods need to conduct optimization from scratch, leading to slow convergence. Transfer learning reuses knowledge from source tasks to boost the performance of current tasks, and thus can be naturally applied to address this issue, i.e., it can utilize

source task data to accelerate the convergence of current target task [2]. The main assumption of transfer learning for BBO is that many real-world problems exhibit certain similarities, and similar tasks often share similar characteristics that can be exploited.

Various transfer learning approaches have been explored concerning the surrogate model, acquisition function, initialization, and search space of BO. Several methods learn all available information from both source and target tasks in a single GP surrogate, and make the data comparable through multi-task GP [37, 16, 38], noisy GP model [31, 25, 13], and deep kernel learning model [47, 11, 10]. Additionally, certain approaches involve training multiple base surrogates and then combining them into a single surrogate [50, 28].

Transfer learning methods on surrogate models, however, may encounter difficulties when scaling to high-dimensional cases, because the influence of the source tasks is easy to diminish progressively with the increase of new observation points [2]. Alternatively, transferring knowledge via the acquisition function can circumvent these issues. Previous research has approached transfer learning for acquisition functions through multi-task BO [37, 20], ensemble GPs [51], and meta-learning strategies such as reinforcement learning-driven strategies [40].

Several approaches consider selecting multiple valuable points for better initial evaluations for the warm-start of optimization. Feurer et al. [7] simply selected the best point from the $t$ most similar tasks as the initialization point. Other works [48, 45] achieved warm-start by constructing a meta-loss, organically combining the mean function of the GP model from source tasks, and using gradient descent to find a set of solutions that minimize the meta-loss. Wistuba and Grabocka [47] tried to find suitable initial points to warm-start BO by minimizing the loss on the source tasks using an evolutionary algorithm.

## 2.3 Search Space Transfer

Compared to the aforementioned methods, search space transfer has many advantages, especially in not limiting the transfer to a specific algorithm component (e.g., acquisition function in BO), but considering the search space that can be used by all BBO algorithms as the transfer object. That is, the learned search space is orthogonal to the optimization process and can be integrated with any advanced optimizer. A well-learned search space can greatly improve optimization efficiency and guide the optimization process to a good result.

The concept of search space transfer was first proposed by Wistuba et al. [49], which defined a region by a center point and a diameter, and pruned away regions deemed less promising. Instead of pruning space, Perrone and Shen [23] considered designing a promising search space for the target task. This approach extracts an optimal solution $x_i^*$ from each task, and employs a simple geometric form (e.g., a box or ellipsoid) to define the smallest subspace encompassing all optimal solutions $x_i^*$. However, it ignores the correlation between tasks, and the space constructed with regular geometry may be too loose for the target task. To address this issue, Li et al. [17] selected the most similar source tasks to the target task in a certain proportion, and used a binary classification method to learn good spaces and bad spaces among these tasks. A voting mechanism is then employed to aggregate information from all selected tasks to determine a newly generated search space for the target task.

These current space transfer methods, however, lack mechanisms to adjust the search space when it is not well-suited for the target task. Furthermore, when source tasks differ significantly from the target task, they tend to devolve into full-space search. The binary nature of evaluating the search space as simply "good" or "bad" is another limitation. Our proposed method will overcome these drawbacks by adapting the search space dynamically according to the similarity between source tasks and the target task, applying MCTS to find a proper subspace, and using a more nuanced and precise numerical evaluation for the goodness of a search subspace.

## 2.4 Monte Carlo Tree Search

MCTS [5] is a search algorithm combining random sampling with a tree search structure, which has been widely applied in the filed of game-playing and decision-making [32, 34]. A tree node represents a particular state in the search space, e.g., stone positions on the board in a GO game. Each tree node has an UCB [1] value during the search procedure, to balance exploration and exploitation.

The UCB of each tree node is calculated by its value $v_m$ and its visit times $n_m$, defined as:

$$\text{ucb}_m = \frac{v_m}{n_m} + 2C_p\sqrt{2\log(n_p)/n_m}, \tag{1}$$

where $C_p$ is a hyper-parameter controlling the balance between exploration and exploitation, and $n_p$ is the visit times of the parent node. In the iterative process of MCTS, a leaf node is systematically selected for expansion, involving four key steps: selection, expansion, simulation, and back-propagation. Initially, the algorithm recursively navigates from the root node to child nodes, prioritizing those with higher UCB values to identify the leaf node $m$. Subsequently, an action is executed based on the state of $m$, leading to the expansion of a new child node (state), $k$. In the simulation step, the node value $v_k$ is determined through random sampling. Finally, through back-propagation, the value and visitation count of the ancestors of $k$ are updated.

MCTS has been used to select important variables automatically for high dimensional BO [35]. LA-MCTS [43] is a scalable BBO algorithm based on learning space partition. Utilizing MCTS, it iteratively divides the search space into small subspaces for optimization. In this framework, the tree's root represents the entire search space denoted as $\Omega$, and each tree node $m$ represents a sub-region $\Omega_m$. The value $v_m$ is determined by the average objective value of the sampled points within the sub-region $\Omega_m$. During each iteration, after selecting a leaf node $m$, LA-MCTS conducts optimization within $\Omega_m$. The sampled points are then employed for clustering and classification, leading to the bifurcation of $\Omega_m$ into two distinct sub-regions: "good" and "bad". These sub-regions are expanded as two child nodes, with the left one denoted as "good" and the right as "bad".

## 3  MCTS-transfer

In this section, we propose a search space transfer learning method based on MCTS, called MCTS-transfer, which can be divided into two major stages: pre-learning stage and optimizing stage. The main idea is to apply MCTS to divide the entire space based on source task data in the pre-learning stage, and adaptively adjust the partition based on newly generated target task data during the optimition process. MCTS-transfer iteratively chooses one partitioned subspace for search and adjusts the partition after sampling each new data point. Assume that there are $K$ source tasks $\{f_i\}_{i=1}^K$ and a target task $T$ that we are going to optimize $\max_{\boldsymbol{x}\in\Omega} f_T(\boldsymbol{x})$, where $\Omega$ is the entire search space. For the $i$-th source task, we have offline dataset $D_i = \{(\boldsymbol{x}_{i,j}, y_{i,j})\}_{j=1}^{|D_i|}$, where $y_{i,j}$ is the observed objective value of $\boldsymbol{x}_{i,j}$, and $|D_i|$ is the number of data points. Let tree $\mathcal{T}$ denotes the search space division process by MCTS.

In the pre-learning stage, $\mathcal{T}$ initially has only one root node, representing the entire search space $\Omega$. The expansion of a tree node $m$ corresponds to the division of the search space $\Omega_m$ that the node $m$ represents. The node expansion follows the rules below. With a set of source task samples $\{(\boldsymbol{x}_i, y_i)\}_{i=1}^n$ in the space $\Omega_m$, we use $k$-means clustering to divide them into two groups. The cluster with a higher average objective value is regarded as the "good" cluster, while the other is the "bad" one. A binary classifier then establishes a decision boundary between the two clusters. In our approach, the space associated with the good cluster becomes the left node, while the space of the bad cluster becomes the right node. We recursively apply this partition process within each node, as shown in Figure 1. The depth of the resulting Monte Carlo tree $\mathcal{T}$ is determined by a threshold hyperparameter $\theta$: a node is divided if it contains more than $\theta$ samples and the contained samples can be clustered into two clusters. The tree $\mathcal{T}$ generated at this stage can give a suitable space partition in advance based on source task data, which serves as a warm-start for the following optimization process. Details of pre-learning will be introduced in Section 3.1.

In the optimization process, we follow the four key steps of MCTS: selection, expansion, simulation, and back-propagation. At each iteration, we select a target node $m$ by tracing the nodes' UCB values. That is, starting from the root, we recursively choose the child node with higher UCB value, until a leaf node $m$ is reached, as Figure 1 displayed. The space $\Omega_m$ represented by $m$ is then considered as a promising search space, where a BO optimizer is used to optimize. The BO optimizer can build a surrogate model using samples in either $\Omega_m$ or $\Omega$. The newly sampled point will be evaluated and used for expanding the node $m$ after updating the clustering in the node. It will further be utilized in back-propagation, where the number of visits and the potential value of each node will be updated. The potential value of a node is calculated by a weighted sum of objective values of the source and target task samples. Note that the tree structure will be reconstructed if there exists a node that the

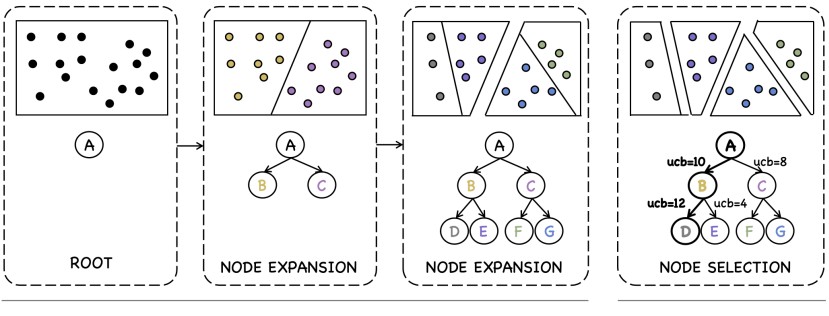

Figure 1: The workflow of MCTS-transfer. In pre-learning stage, MCTS-transfer builds the tree by clustering and classifying the samples apart recursively, until all nodes are not splitable. In optimization stage, the initial search space is based on the pre-learned tree. We will trace child node with greater UCB from ROOT and find the target leaf node to do optimization.

potential value of its right child exceeds that of the left one, violating the rule of our tree construction and implying that the current structure is not good. Please see Sections 3.2 and 3.3 for the details of node potential update and tree structure reconstruction.

Unlike LA-MCTS, MCTS-transfer not only considers the information of the target task when calculating node values but also takes into account the information from source tasks, enabling it to be used for search space transfer. Additionally, we propose adaptive weights for source tasks and validate their effectiveness through our experiments.

## 3.1 Search Space Pre-Learning

Compared to standard BBO algorithms, which sample randomly across the entire space at the beginning, our method leverages information from source tasks to concentrate sampling within a generally "good" space, providing a warm-start initialization.

In the pre-learning stage, tree $\mathcal{T}$ initially has only one node, i.e., ROOT node. All source task samples are collected in this node, recursively clustered and classified, leading to node expansion. When none of the leaf nodes can be further bifurcated, i.e., contains more than $\theta$ samples and can be clustered into two clusters, $\mathcal{T}$ is finally formed. In this process, we keep the rule that the left node is "better" (i.e., has a larger potential value) than the right node, so that we can easily identify that the leftmost leaf is the best and the rightmost leaf is the worst. The potential value of a node $m$ in the pre-learning stage is estimated by the average $y_{i,j}$ of all source task points within it, i.e.,

$$p_m = \frac{\sum_{i \leq K} \sum_{(\boldsymbol{x}_{i,j}, y_{i,j}) \in D_i \cap \Omega_m} y_{i,j}}{\sum_{i \leq K} |D_i \cap \Omega_m|}. \tag{2}$$

This tree embed historically good regions into specific tree nodes. Intuitively, the spaces represented by left leaves have higher potential of being good spaces than right ones.

The first iteration of MCTS-transfer will utilize the resulting tree $\mathcal{T}$ generated in the pre-learning stage. Starting from the root node, it recursively selects nodes with higher UCB values until reaching a leaf node. The UCB value is calculated as Eq. (1), where $v_m$ is replaced by the potential value $p_m$ in Eq. (2). In MCTS-transfer, $n_m$ and $n_p$ represent the number of samples, including those from source and target tasks, in the subspace $\Omega_m$ and the parent subspace of $\Omega_m$, respectively. Consequently, the algorithm preferentially samples from those regions that have shown to yield favorable outcomes in the source tasks.

## 3.2 Node Potential Update

In each iteration of MCTS-transfer, after evaluating a new sample, we use it to update the similarity between the source tasks and the target task, and then update each node potential. Note that the potential value $p_m$ used in the pre-learning stage only contains information from source tasks. But in

the optimizing stage, the calculation of $p_m$ includes both information from the source tasks $\{D_i\}_{i=1}^K$ and the target task $D_T = \{(\boldsymbol{x}_{T,j}, y_{T,j})\}_{j=1}^t$, and considers the similarity between them, i.e.,

$$p_m = \gamma^{t-1} \frac{\sum_{i \leq K} w_i \overline{y}_{i,m}}{\sum_{i \leq K} w_i} + \overline{y}_{T,m}, \tag{3}$$

where $\overline{y}_{i,m} = \sum_{(\boldsymbol{x}_{i,j}, y_{i,j}) \in D_i \cap \Omega_m} y_{i,j} / |D_i \cap \Omega_m|$, and $\overline{y}_{T,m} = \sum_{(\boldsymbol{x}_{T,j}, y_{T,j}) \in D_T \cap \Omega_m} y_{T,j} / |D_T \cap \Omega_m|$, which are the average objective values of the samples of $D_i$ and $D_T$ in $\Omega_m$, respectively. There are two important parameters in Eq. (3).

- $\gamma$: A decay factor, which is used to adjust the overall influence of source tasks throughout the optimization process. A smaller $\gamma$ accelerates the forgetting of the source task data.

- $w_i$: A weighting factor, which reflects the influence of the $i$-th source task, and is determined by the similarity between the $i$-th source task and the target task. A larger weight represents higher similarity and implies a greater influence of the $i$-th source task's samples on the potential calculation of a tree node.

To calculate $w_i$, we measure the distance $Distance(D_i, D_T)$ between the $i$-th source task and the target task by $Distance(\boldsymbol{x}_i^*, \boldsymbol{x}_T^*)$, where $\boldsymbol{x}_i^*$ and $\boldsymbol{x}_T^*$ denote the mean of the best $N$ sampled points of these two tasks, respectively. The source tasks are ranked according to their distances to the target task. A smaller rank implies a smaller distance, i.e., a higher similarity. The weight $w_i$ is then calculated as

$$w_i = \begin{cases} 1.0 - \frac{r_i}{\alpha N_m} & \text{if } r_i < \alpha N_m \\ 0.1 & \text{otherwise} \end{cases}, \tag{4}$$

where $r_i$ is the rank of the $i$-th source task, $N_m$ is the number of source tasks that have solutions located in $\Omega_m$, and $\alpha \in [0, 1]$. After sampling a new data point in each iteration of MCTS-transfer, the ranks and weights are updated accordingly, which are then used to update the potential value of each node. We also consider other ways of calculating distances and weights, which are introduced and empirically compared in Appendix C.

### 3.3 Tree Structure Reconstruction

When constructing the search tree, the left child of a node is always better than the right child, i.e., the potential of the left child always exceeds that of the right child. However, after sampling a new data point and updating the potential of each node in each iteration of MCTS-transfer, this property may be violated. If this happens, it indicates that the current space partition is not ideal and needs adjustment. Specifically, we backtrack from the problematic leaf nodes to the highest-level ancestor node that upholds the desired property, and then proceed to reconstruct the subtree from that ancestor node. The subtree reconstruction process is consistent with the process of node expansion.

The detailed process is presented in Algorithm 2. We apply breadth-first search to traverse all tree nodes and use a queue $\mathcal{Q}$ to manage the sequence of nodes. In addition, we set a queue $\mathcal{N}$ to store the nodes that need to be reconstructed. If the potential of the right child of a node is better than that of the left child (line 6), the subtree of this node is deleted (line 7), and it should be reconstructed and is added into the queue $\mathcal{N}$ (line 8). Otherwise, the two child nodes will enter into the queue $\mathcal{Q}$ (lines 10–11) and will be examined later. After traversing the tree $\mathcal{T}$, we reconstruct the subtrees of nodes in $\mathcal{N}$ (lines 15–19). If a node in $\mathcal{N}$ is splitable, i.e., the contained samples in the node exceeds $\theta$ and can be clustered and divided by a binary classifier, it will be further expanded. Thus, Treeify can fine-tune the tree structure and reserve some history information; meanwhile, it can adaptively be more suitable to the target task.

### 3.4 Details of MCTS-transfer

The detailed procedure of MCTS-transfer is presented in Algorithm 1. It begins with a pre-learnned MCTS model $\mathcal{T}$ based on source task data (line 1), and initializes an empty set $D_T$ to store samples of the target task (line 2). In each iteration, it selects a leaf node $m$ based on the UCB value (line 4). If the target task already has samples in the space $\Omega_m$ represented by $m$, the algorithm conducts BO within $\Omega_m$ and gets a candidate point $\boldsymbol{x}_{T,t}$ (lines 8–9); otherwise, the candidate point is randomly selected in $\Omega_m$ (line 6). The sampled point $\boldsymbol{x}_{T,t}$ is then evaluated and added into $D_T$ (line 11). To

**Algorithm 1** MCTS-transfer

---

**Input**: Search space $\Omega$, objective function $f_T(\cdot)$, data $\{D_i\}_{i=1}^K$ of $K$ source tasks, and number $N$ of objective evaluations

1:  Initialize a MCTS model $\mathcal{T}$ by search space pre-learning as introduced in Section 3.1;
2:  $D_T = \emptyset$;
3:  **for** $t = 1, 2, \ldots, N$ **do**
4:      Select a leaf node $m$ by UCB-based selection;
5:      **if** $|D_T| = 0$ **then**
6:          Select a candidate point $\boldsymbol{x}_{T,t}$ randomly in $\Omega_m$
7:      **else**
8:          Train a GP model on $D_T$;
9:          Select a candidate $\boldsymbol{x}_{T,t}$ in $\Omega_m$ by optimizing an acquisition function
10:     **end if**
11:     Evaluate $\boldsymbol{x}_{T,t}$ to get $y_{T,t}$, and let $D_T = D_T \cup \{(\boldsymbol{x}_{T,t}, y_{T,t})\}$;
12:     Calculate $Distance(D_i, D_T)$, and sort them in ascending order to get the ranks $r_i$;
13:     For each source task $f_i$, update its weight $w_i$ by Eq. (4), where $i = 1, \cdots, K$;
14:     For each node $n$, update its potential value $p_n$ by Eq. (3);
15:     **if** $m$.isSplitable **then**
16:         Expand $m$
17:     **end if**
18:     Back-propagate and update the visit times and samples of nodes on the path to $m$;
19:     $\mathcal{T} \leftarrow$ **Treeify** $(\mathcal{T})$
20: **end for**

---

make full use of the information from the source tasks, we take the similarity between a source task and the target task into account, and try to assign higher weights to more similar source tasks, as introduced in Section 3.2. With new samples added in $D_T$, the distance between each source task $D_i$ and the target task $D_T$ may change. Thus, the distance, rank and weight of each source task are re-calculated in lines 12–13 according to Eq. (4), and the potential value of each node in the tree is re-evaluated in line 14 according to Eq. (3). After that, the node $m$ is expanded in lines 15–17 if it is splitale, i.e., contains more than $\theta$ samples of the target task and can be clustered into two clusters. Note that for node expansion, only samples of the target task are considered. In line 18, the back-propagation step is performed to refresh the information of each node on the path to $m$, including the visit times and the contained samples. Finally, we will check whether the updated tree conforms to the property that the left child of a node has a larger potential value than its right node. For any node violating this property, its sub-tree will be reconstructed. This process is accomplished by employing the Treeify procedure in Algorithm 2 in Appendix A.

In MCTS-transfer, the tree structure exhibits several properties that align with the requirements of search space transfer learning. 1) Tree structure is natural to model the search space, where the root represents the entire search space, and the node expansion corresponds to the space partition with each child node representing a subspace. 2) The node potential evaluation allows for the extraction of multiple irregularly shaped promising search subspaces represented by multiple leaf nodes. 3) The tree can be adjusted by updating and expanding nodes on the basis of the original tree, so that new information can be absorbed while historical information can also be retained, i.e., the tree structure is inheritable. This enables the knowledge of space partition from source task data to be transferred to the target task.

## 4 Experiments

In this section, we introduce a simple case to highlight the features of existing search space transfer algorithms and assess the effectiveness of MCTS-transfer. We also conduct experiments across a variety of tasks, including the synthetic benchmark BBOB, real-world problems, Design-Bench and the hyper-parameter optimization benchmark HPOB. The details of the problems, data and experimental results can be seen in Appendix B.2, B.3 and E.

**Compared methods.** Our baselines consist of two non-transfer BO, i.e., basic GP, and LA-MCTS [43], three search space transfer algorithms, i.e., Box-GP [23], Ellipsoid-GP [23], Supervised-

GP [17]), and one state-of-the-art surrogate model transfer BO algorithm PFN [21]. Detailed configurations and settings of the hyper-parameters for each algorithm are provided in Appendix B.1.

**Basic Settings.** In our experimental settings, we explore two types of transfer learning—similar and mixed transfer—to demonstrate the advantages of MCTS-transfer in handling complex source tasks. Similar transfer involves using data from similar source tasks, while mixed transfer uses a combination of similar and dissimilar source task data for pre-training. Given the challenges in real-world scenarios of determining task similarity and selecting appropriate tasks, our focus will be on evaluating MCTS-transfer's performance in the more complex scenario, i.e., mixed transfer.

### 4.1 Motivating Cases: Sphere2D

We first conduct a simple but important experiment on Sphere2D to verify our motivation and evaluate the performance of existing search space transfer methods. The Sphere2D function is defined as $f(\boldsymbol{x}) = (\boldsymbol{x} - \boldsymbol{x}^*)^2$, with $\boldsymbol{x}^*$ representing the optimal solution. We generate three source task datasets $D_{(5,5)}$, $D_{(5,-5)}$, and $D_{(-5,-5)}$, each containing 100 samples, by assigning $(5,5)$, $(5,-5)$, and $(-5,-5)$ to $\boldsymbol{x}^*$ and sampled by standard BO. The target task is defined as $f(\boldsymbol{x})$ with $\boldsymbol{x}^* = (4,4)$.

Unlike the basic setting, we use the mixed setting and the dissimilar setting. Dissimilar setting uses only $D_{(5,-5)}$ and $D_{(-5,-5)}$ for pre-training, which are most dissimilar to the target task. This setting is to test MCTS-transfer's ability to handle extreme cases without similar source tasks.

MCTS-transfer-GP is compared against three search space transfer algorithms—Box-GP, Ellipsoid-GP, and Supervised-GP—in both mixed and dissimilar settings, as shown in Figure 2. Box-GP and Ellipsoid-GP ignore task similarity and limit the search space to geometric areas encompassing all source task optima. In the dissimilar transfer setting, these methods fail because the target task's optimal solution $\boldsymbol{x}^* = (4,4)$ lies outside the defined subspace of source task optima. Supervised-GP takes task similarity to guide the selection and combination of promising region with a voting system. Although it pays attention to the importance of task similarity and can be applicable in dissimilar transfer, it tends to be an entire-space optimization algorithm when source tasks exhibit high variance or are dissimilar to the target task. MCTS-transfer recursively divides the search space, dynamically ajusts task weights and evaluates the subspace potential based on the task similarity. In addition, the method tends to increasingly rely on target task data due to decay factor $\gamma$. Therefore, MCTS-transfer achieves successful convergence with low variance in mixed and dissimilar transfer. The right part of Figure 2 displays the weight change curves for the three source tasks, confirming the anticipated results. Higher weights are assigned to datasets $D_{(5,5)}$ and $D_{(-5,5)}$, indicating that our method effectively recognizes and prioritizes tasks more similar to the target task.

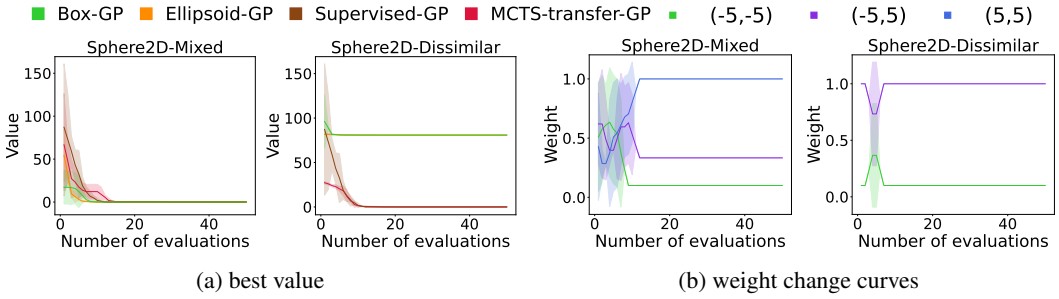

Figure 2: Comparison of MCTS-transfer and other algorithms on Sphere2D

### 4.2 Synthetic Functions

BBOB [9] is a popular benchmark for BBO. It offers 24 synthetic functions tailored for the continuous domain. We randomly select one function from each of the five function classes of BBOB as our target task, i.e., GriewankRosenbrock, Lunacek, Rastrigin, Rosenbrock, and SharpRidge.

As shown in Figure 3a, compared with LA-MCTS, we find that the results can indeed be greatly improved after space transfer, proving the effectiveness of MCTS-transfer. In mixed transfer, MCTS-transfer can still maintain warm-start, thanks to its ability to provide multiple compact promising

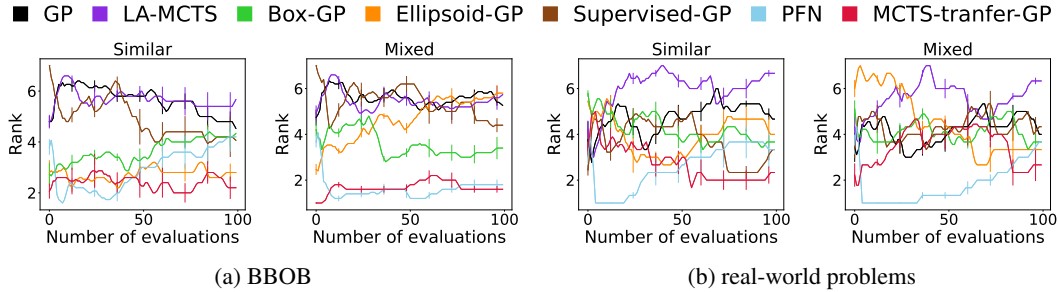

Figure 3: Comparison of MCTS-transfer and other algorithms on BBOB and real-world problems

subspaces by effective search space pre-learning. Throughout the optimization process, MCTS-transfer, combined with a simple GP, is able to reach or even surpass SOTA's surrogate model transfer algorithm PFN, reflecting the superior performance.

### 4.3 Real-world Problems

To demonstrate the practical value of MCTS-transfer, we test MCTS-transfer with three real-world problems, including LunarLander, RobotPush, and Rover.

Figure 3b shows comparison on similar and mixed transfer. Based on the overall ranking, MCTS-transfer-GP outperforms other search space transfer baselines.In mixed transfer, although it's surpassed by PFN initially, MCTS-transfer-GP is finally able to find better solutions even if there are dissimilar tasks misleading. The ability to adaptively correct the search space may come from the efficient utilization of source task data by reasonable node potential evaluation, node expansion, and tree reconstruction.

### 4.4 Design-Bench Problems

We further verify the performance of MCTS-transfer in more complex and high-dimensional problems from Design-Bench [39], including three continuous problems, Superconductor, Ant morphology and D'Kitty Morphology.

In similar and mixed transfer, MCTS-transfer-GP gets the best ranking after 40 iterations. The stable performance indicates that the algorithm can effectively transfer and utilize previous experience when facing different source tasks. This ability is particularly important for black-box problems because it allows the system to quickly adapt to unseen situations while maintaining high performance.

### 4.5 Hyper-parameter Optimization

Hyper-parameter optimization(HPO) is a common application of BBO transfer learning. The HPO-B benchmark [24], sourced from OpenML, includes 176 search spaces/algorithms and 196 datasets, totaling 6.4 million HPO evaluations. Details of Source task data selection and curves of each problem are placed in Appendix B.3 and E, respectively.

The HPOB benchmark has several challenges. Firstly, for these problems, algorithms are easy to converge on a certain area, leading to rapid convergence. Secondly, the source tasks are complex, presenting significant variability in data distribution and scale. To tackle the difficulties, the search algorithm should be equipped with ability to handle complex source tasks and robust optimization capabilities.

Experimental results are shown in Figure 4b, which represent the mean ranks and the variance of all algorithms on 39 test problems. In both transfer settings, the baseline ranks are very close, because there are only tiny gaps among final convergence values, which can be seen in Appendix E. However, our method still have clear advantages. Although our method doesn't have enough warm-start strength when the source tasks are diverse and complex, but it can still adjust the search tree structure and locate the better search area, exceeding existing search space transfer algorithms.

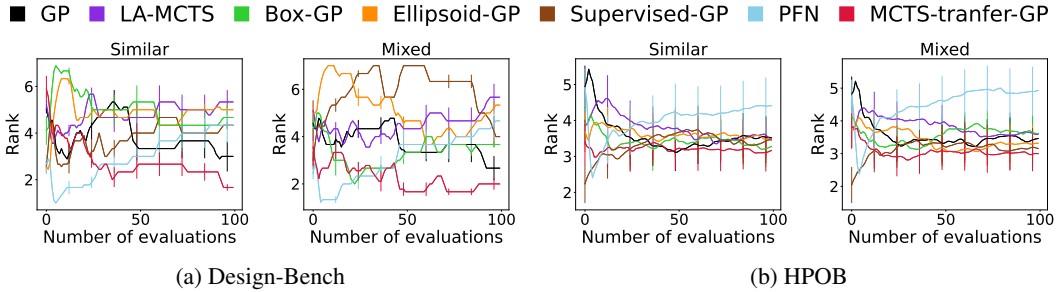

(a) Design-Bench             (b) HPOB

Figure 4: Comparison of MCTS-transfer and other algorithms on Design-Bench and HPOB

## 4.6 Runtime Analysis

To verify whether MCTS-transfer incurs excessive time overhead, we analyze the runtime proportion of each component of MCTS-transfer on BBOB and three real-world tasks (i.e., LunarLander, RobotPush, and Rover). We divide MCTS-transfer into three main components: evaluation, backpropogation and reconstruction. The evaluation time includes the time required for surrogate model fitting, candidate solution selection and evaluation, which is the common component shared by all optimization algorithms. Backpropagation and reconstruction constitute the two principal modules specific to MCTS-transfer.

As shown in Figure 5, the additional computational burden introduced by MCTS-transfer (i.e., backpropagation and reconstruction) represents a relatively minor fraction of the total runtime, particularly in the three expensive real-world problems. In expensive evaluation problems, MCTS-transfer can bring great improvement by introducing small additional computational overhead. Other details, such as the exact time cost of each component and the frequency MCTS subtree reconstruction, are provided in Appendix F.

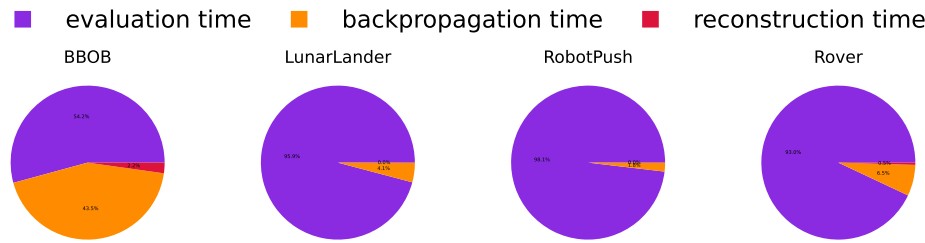

Figure 5: Time cost proportion of evaluation, backpropogation and reconstruction of MCTS-transfer

## 5 Conclusions and Limitations

In this paper, we propose MCTS-transfer to solve the expensive BBO problem. MCTS-transfer uses MCTS to perform search space transfer, extracting similar tasks during the optimization process and giving more accurate space partition results. Compared with other space transfer algorithms, our algorithm is more generalizable. Specifically, it can extract the most similar source tasks and give them higher weights to accelerate optimization. Besides, it is reliable because it can dynamically adjust the tree structure, improving the probability that the optimal solution falls in search space of the chosen node. Comprehensive experiments on BBOB, real-world problems, Design-Bench and HPOB demonstrate the effectiveness of our algorithm. However, there are still limitations, such as the inability of MCTS-transfer to handle search space transfer tasks for problems with different domains or different dimensions. Future work will focus on solving the heterogeneous space transfer [6], exploring more accurate similarity measures and trying to give the theoretical analysis of the effectiveness of MCTS-transfer.

## Acknowledgments and Disclosure of Funding

We thank the reviewers for their insightful and valuable comments. This work was supported by the Science and Technology Project of the State Grid Corporation of China (5700-202440332A-2-1-ZX).

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

# A  Treeify Pseudocode

To maintain the property that the potential of the left child should exceed the potential of the right child for each node, tree reconstruction should be implemented in each iteration. The process of adjusting the tree structure, i.e., deleting and reconstructing subtrees that violate the desired property, is called Treeify. The pseudocode is provided below, as shown in Algorithm 2.

---

**Algorithm 2** Treeify

---

**Input**: a MCTS model $\mathcal{T}$
**Output**: an updated MCTS model $\mathcal{T}$

 1: Initialize empty queues $\mathcal{Q}, \mathcal{N}$;
 2: $\mathcal{Q} \leftarrow \mathcal{Q}$.Enqueue(ROOT of $\mathcal{T}$);
 3: **while** not $\mathcal{Q}$.isEmpty **do**
 4:     $node \leftarrow \mathcal{Q}$.Dequeue;
 5:     **if** not $node$.isLeaf **then**
 6:         **if** $p_{node.left} < p_{node.right}$ **then**
 7:             Delete the subtree of $node$;
 8:             $\mathcal{N} \leftarrow \mathcal{N}$.Enqueue($node$)
 9:         **else**
10:             $\mathcal{Q} \leftarrow \mathcal{Q}$.Enqueue($node.left$);
11:             $\mathcal{Q} \leftarrow \mathcal{Q}$.Enqueue($node.right$)
12:         **end if**
13:     **end if**
14: **end while**
15: **for** each $node$ in $\mathcal{N}$ **do**
16:     **if** $node$.isSplitable **then**
17:         Expand $node$
18:     **end if**
19: **end for**

---

# B  Detailed settings

## B.1  Algorithms

We use the authors' reference implementations for LA-MCTS[3], Box-GP[4], Ellipsoid-GP[5], Supervised-GP[6] and PFNs4BO[7]. The hyper-parameters of all compared algorithms are summarized as follows:

- **GP-EI**. We use the GP model in Scikit-learn[8] and the ExpectedImprovement acquisition [46]. The kernel is set to be ConstantKernel(1.0)*Matern(length_scale=1.0, nu=2.5). To optimize the acquisition function, we generate 10000 points randomly and then select the one with the maximum expected improvement to be evaluated, which is similar to LA-MCTS [43].

- **LA-MCTS** [43]. We set $C_p = 0.1, \theta = 10$. We adopt global GP as the modeling approach. SVM with rbf kernel is used for space division in Sphere2D, BBOB, and HPOB, while Logistic Regression is applied for real-world problems. In the sampling process, we sample the entire space 10,000 times, retaining qualifying candidate points, and repeat this process three times. The parameters of GP are consistent with GP-EI. All other parameters are set to their default values.

- **Supervised-GP**. The sampling process is consistent with LA-MCTS. The parameters of GP are consistent with GP-EI. All other parameters are set to their default values.

---

[3] https://github.com/facebookresearch/LaMCTS
[4] https://github.com/PKU-DAIR/SSD/tree/master
[5] https://github.com/PKU-DAIR/SSD/tree/master
[6] https://github.com/PKU-DAIR/SSD/tree/master
[7] https://github.com/automl/pfns4bo
[8] https://github.com/scikit-learn/scikit-learn

- **Box-GP**. The sampling process is consistent with LA-MCTS. The parameters of GP are consistent with GP-EI. All other parameters are set to their default values.

- **Ellipsoid-GP**. The sampling process is consistent with LA-MCTS. The parameters of GP are consistent with GP-EI. All other parameters are set to their default values.

- **MCTS-transfer-GP**. We set $\gamma = 0.99, \alpha = 0.5$. We adopt 5 best solutions distance as similarity measure and linear-change strategy as weight assignment. Other parameters and sampling method is consistent with LA-MCTS. The parameters of GP are consistent with GP-EI.

- **PFN**. We set epochs = 5000, batch size=256, learning rate=3e-4, warm up steps = 100, model dimension = 256, head = 4, layer number = 4, hidden layer size = 256, dropout=0.1.

## B.2 Problems

We consider the following three real-world problems in our experiments.

**LunarLander.**[9] This problem is to learn the parameters of a controller for a lunar lander, which is implemented in OpenAI gym.[10] It involves a 12-dimensional continuous input space depicting the lander's actions. Our goal is to enhance the control algorithm to maximize the mean terminal reward across a consistent batch of 50 randomly generated landscapes, incorporating varying initial positions and velocities.

**RobotPush.**[11] This function computes the distance between a predefined target location and two objects that are manipulated by a pair of robotic appendages. The movement trajectory of these objects is governed by a set of 14 parameters, which encapsulate attributes such as position, orientation, speed, and direction of motion. We need to control the robot to push items to a designated location. This is implemented with a physics engine Box2D[12].

**Rover.**[13] The task optimizes 2D trajectories for a rover by defining start and goal positions and a cost function over the state space [44]. The trajectory costs $c(x)$ is computed for solutions $x$ within a 60-dimensional unit hypercube. We need to design a reasonable trajectory to minimize the cost.

To verify the performance of MCTS-tranfer in more complex and high-dimensional cases, we consider the 3 continuous problems from Design-Bench[14] [39].

**Superconductor.** It's a critical temperature maximization for superconducting materials. This task is taken from the domain of materials science, where the goal is to design the chemical formula for a superconducting material that has a high critical temperature. The search space is a continuous space with 86 dimensions.

**Ant morphology.** It's a robot morphology optimization. The goal is to optimize the morphological structure of Ant from OpenAI Gym[15] to make this quadruped robot to run as fast as possible. The search space is a continuous space with 60 dimensions.

**D'Kitty Morphology.** It's robot morphology optimization. The goal is to optimize the morphology of D'Kitty robot to navigate the robot to a fixed location. The search space is a continuous space with 56 dimensions.

## B.3 Source Task Data Construction for Similar and Mixed Transfer

**Data collection** The source task data for pre-training is generated by seven optimization algorithms: Random Search [3], Shuffled Grid Search, Hill Climbing, Regularized Evolution [27], Eagle Strategy [54], and GP-EI [12]. These methods span heuristic, evolutionary, and BO techniques, providing a diverse set of optimization behaviors for our analysis.

---

[9]`https://github.com/yucenli/bnn-bo/blob/main/test_functions/lunar_lander.py`

[10]`https://gym.openai.com/envs/LunarLander-v2`

[11]`https://github.com/zi-w/Ensemble-Bayesian-Optimization/blob/master/test_functions/push_function.py`

[12]`https://box2d.org/`

[13]`https://github.com/zi-w/Ensemble-Bayesian-Optimization/blob/master/test_functions/rover_function.py`

[14]`https://github.com/brandontrabucco/Design-Bench`

[15]`https://github.com/openai/gym`

**BBOB** In order to obtain similar source task data of function $f$, we perform some similar transformations on $f$ to obtain function family $\mathcal{F}_f$ with seed from 0 to 499. Then we use the sampling algorithms mentioned above with randomly selected seed to collect 300 samples on each function in $\mathcal{F}_f$. In this section, we randomly select 20 datasets from datasets of $\mathcal{F}_f$ for function $f$. In the similar setting, We only use selected datasets of $\mathcal{F}_f$. In mixed setting, we use selected datasets of all 5 synthetic functions.

**Real-world problems and Design-Bench problems** Since it is difficult to find data with the same dimensions in real-world problems, we choose to use the artificial function $Sphere$ of the same dimension to generate similar data and dissimilar data and mix it into the source task data set. Among them, to simulate similar functions, we set the optimal point $\boldsymbol{x}^*$ of the sphere function to be the optimal point of all these algorithms found so far; the optimal point of the $Sphere$ function that simulates dissimilar functions is set to $1 - \boldsymbol{x}^*$. In the similar setting, we randomly select 20 trajectories from the problem with different seeds and function transformations, combined with data generated by similar sphere function, as the dataset. In the mixed setting, we further add 7 trajectories from dissimilar sphere function. In this section, it is a problem with significantly more similar source tasks than dissimilar tasks.

**HPOB** We choose search spaces from HPO-B-v3 and divide them into 4 groups based on dimensions. Specailly, we obtain group 5860 and 5970 with dimension 2, group 5859 and 5889 with dimension 5, group 7607 and 7609 with dimension 9, group 5806 and 5971 with dimension 16. In the similar setting, we only use the training data from the search space itself for pre-learning. In the mixed setting, we use all training data of search spaces in the same group.

## C Discussions of methods

### C.1 Task Similarity

Measuring the similarity of two tasks based on the dataset of a source task $D_i$ and the dataset of the target task $D_T$ is a complex task. We define task distance as a measure of similarity. Smaller distances between tasks represent greater similarity. Here we mainly focus on two types of methods: point-based similarity measures and distribution-based similarity measures. The former includes optimal solutions distance, best $N$ solutions mean distance, best $N$ percent solutions mean distance; and the latter includes Kendall coefficient of datasets, Kullback-Leibler Divergence(KL Divergence) of distributions. Detailed description are as follows:

**Optimal solutions distance** Let $\boldsymbol{x}_T^*$ be the best solution of target task, and $\boldsymbol{x}_i^*$ be the best solution of target task. The task distance is calculated as $Distance(\boldsymbol{x}_T^*, \boldsymbol{x}_i^*)$.

**Best N (or N percent) solutions mean distance** Similar to optimal solutions distance, we replace $\boldsymbol{x}_i^*$ and $\boldsymbol{x}_i^*$ with the mean of best N (or N percent) of target task and source task $i$, denoted as $\bar{\boldsymbol{x}}_T^*$ and $\bar{\boldsymbol{x}}_i^*$. The task distance is $Distance(\bar{\boldsymbol{x}}_T^*, \bar{\boldsymbol{x}}_i^*)$. Intuitively, it's more robust than directly using the best solution.

**Kendall coefficient** Kendall coefficient is a measure of rank correlation, which can be used to evaluate the consistency of two datasets. We first build a surrogate model (usually GP) $M_i$ on $D_i$, and then predict all sample values in $D_T$. According the ranking results of all these data, we get the similarity of the two tasks by Eq (5), where $\mathbb{I}$ is the indicator function and $\oplus$ is the exclusive-nor operation, in which the statement value is true only if the two sub-statements return the same value.

$$Distance(D_T, D_i) = \frac{2 \sum_{j=1}^{|D_T|} \sum_{k=j+1}^{|D_T|} \mathbb{I}((M_i(x_{T,j}) < M_i(x_{T,k})) \oplus (y_{T,j} < y_{T,k}))}{|D_T| * (|D_T| - 1)} \quad (5)$$

**KL divergence of distribution** KL divergence is a measure of the dissimilarity between two probability distributions, usually expressed as Eq. (6). For $D_T$ and $D_i$, we first use Guassian KDE to estimate their density of distribition, denoted as $p$ and $q$. Then we sample a set of $\{\boldsymbol{x}_i\}_{i=1}^n$ in $\Omega$ and

evaluate them. KL divergence is calculated based on $\{p(x)\}_{i=1}^{n}$ and $\{(q(x)\}_{i=1}^{n}$.

$$Distance(D_T, D_i) = KL(p\|q) = \sum_x p(x) \log \frac{p(x)}{q(x)} \tag{6}$$

### C.2    Weight Assignment

Based on the similarity between the source tasks and the target task, we can rank the source tasks, and the smaller rank $r_i$ means higher weight assigned. We can choose different weight change strategies, for example, Linear Change Strategy in Eq. (7) , Exponential Change Strategy in Eq. (8) or All One Strategy, i.e., all source task weights are set to be 1, to control the proportion of influencial source tasks.

$$w_i = \begin{cases} 1.0 - \frac{r_i}{\alpha N_m} & \text{if } r_i < \alpha N_m \\ 0.1 & \text{otherwise} \end{cases}, \tag{7}$$

$$w_i = \beta^{r_i}, 0 \le r_i \le N_m - 1 \tag{8}$$

In Eq. (7), $N_m$ is the number of source tasks that have solutions located in $\Omega_m$, and $\alpha \in [0, 1]$ controls the proportion. In Eq. (8), $\beta \in [0, 1]$ is a hyper-parameter of weight change speed.

### C.3    Discussion on Conditional search space.

MCTS-transfer can also be applied in conditional search space, which is common in hyper-parameter optimization. For conditional optimization, we consider the problem $\min_{\boldsymbol{x} \in \mathcal{X} \subset \mathbb{R}^d} f(\boldsymbol{x})$. Specifically, the search space is tree-structured, formulated as $\mathcal{T} = \{V, E\}$, where $v \in V$ is a node representing subspace and $e \in E$ is an edge representing condition. The objective function is also defined based on $\mathcal{T}$, formulated as $f_{\mathcal{T}}(\boldsymbol{x}) := f_{p_j, \mathcal{T}}(\boldsymbol{x}|l_j)$, where $p_j$ is a condition and $\boldsymbol{x}|l_j$ is the restriction of $\boldsymbol{x}$ to $l_j$ [18]. In the pre-learning stage, it builds subtrees for each $v \in V$ and generates the MCTS model $\mathcal{T}'$ based on $\mathcal{T}$. In each iteration, followed by UCB value, it finds the target node $m$ located in the subtree of $v$ with condition $p_i$, optimizes in $\Omega_m$, selects and evaluates the candidate using $f_{p_i, \mathcal{T}}(\boldsymbol{x}|l_i)$. After that, it updates the task weights and node potential in the whole tree $\mathcal{T}'$ and tries to reconstruct the tree. Note that the tree reconstruction only happens in each subtree of $v \in V$.

### C.4    Discussion between the state value in MCTS of AlphaZero and MCTS-Transfer.

MCTS measures state value differently in reinforcement learning and black box optimization. For example, AlphaZero [33] is designed to master complex games through self-play without relying on human knowledge or guidance. MCTS plays a crucial role in AlphaZero's decision-making process, whose state value is used to predict the expected future reward from the current state to the end, rather than the historical information. Different from that a state's future reward of AlphaZero can be obtained through multi-step simulations, i.e., alternating decisions through self-play, evaluation values in BBO can only be obtained through actual evaluations. Consequently, MCTS-transfer utilizes historical information. We have observed that some recent look-ahead BO works [52, 53] have been used to predict the expected value of future steps in BBO problems, which have the potential to be applied in MCTS-transfer as estimates for state values to further improve the performance.

### C.5    Discussion of combining MCTS-transfer with other advanced algorithms

MCTS-transfer is a general search space transfer learning framework, which can be combined with other advanced algorithms. We equip MCTS-transfer with PFN and test MCTS-transfer-PFN on mixed real-world problems. As shown in Figure 6, MCTS-transfer-PFN makes further improvements compared to MCTS-transfer-GP and PFN after around 80 iterations, and its ranking is stable and excellent throughout the optimization. This result demonstrates the versatility of MCTS-transfer combining with advanced BO algorithms.

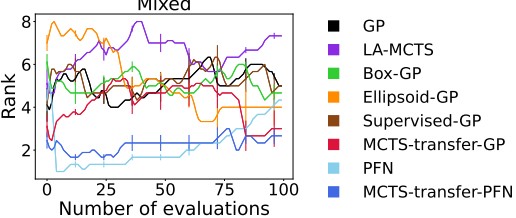

Figure 6: Experimental results on mixed real world problems (with MCTS-transfer-PFN)

# D Sensitivity Analysis of Hyper-parameters of MCTS-transfer

Main hyper-parameters of MCTS-transfer includes modeling approach, similarity measures, weight assignment strategies, decay factor $\gamma$, important source task ratio $\alpha$, exploration factor $C_p$, spliting threshold $\theta$ and the binary classifier. We conduct sensitivity analysis of these important parameters on real-world problems LunarLander, RobotPush, Rover under mixed setting.

**Modeling approach**    In MCTS-transfer, we can choose to train GP on full $D_T$ dataset or on the selected points in $\Omega_m$. As shown in Figure 7, the global GP can utilize more global information, demonstrating enhanced search capabilities in these three problems.

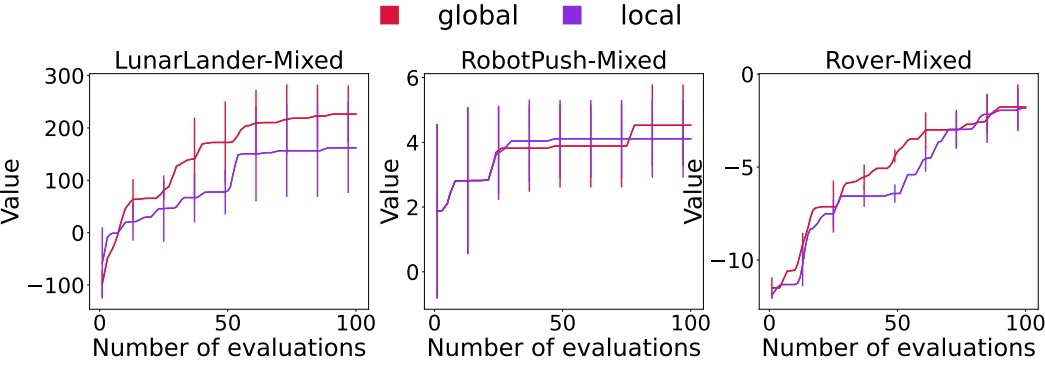

Figure 7: Sensitivity Analysis of Local and Global Modeling Approaches

**Similarity measures**    We propose 5 similarity measures in this paper, including optimal solutions distance, best N or N percent solutions mean distance, Kendall coefficient and KL divergence, as C.1 displayed. The first three methods are point-based measures and the last three are distribution-based methods. Here we set N=5 and N=30% separately for best N solutions and best N percent solutions, and use Gaussian KDE to evaluate KL divergence of distributions. The results can be seen in Figure 8. The distribution-based methods give more precise measure of similarity, but the point-based methods are designed to focus on the optimal region. Each problem has its own suitable measure of similarity. In mixed transfer real-world problems, KL divergence is more appropriate to the problem.

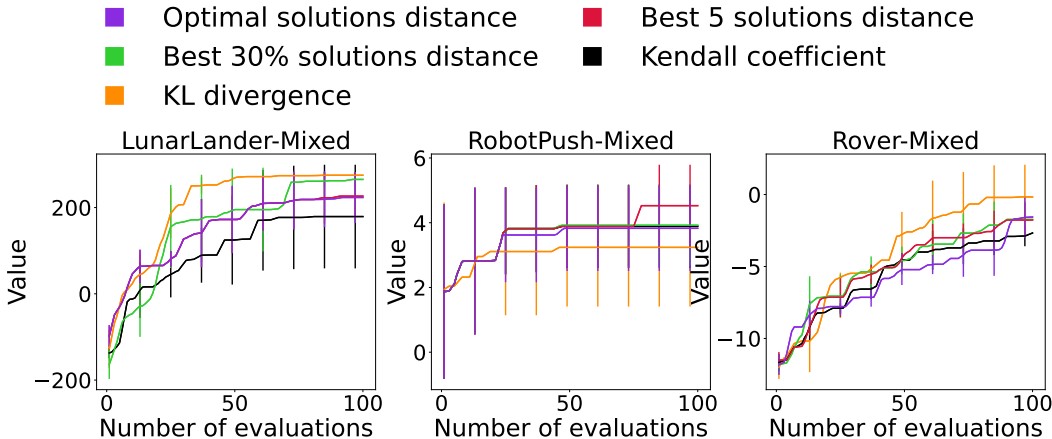

Figure 8: Sensitivity Analysis of Similarity Measures

**Weight assignment**    To assign weights to source tasks, we consider three strategies: linear-change strategy(i.e., Eq. (7)), the exponential-change strategy (i.e., Eq. (8)) and all-one strategy. We set $\alpha = 0.5$ for linear-change strategy, $\beta = 0.5$ for exponential-change strategy. As shown in Figure 10,

the exponential-change strategy, since the weight decaying rapidly with the rank, can only effectively leverages the 1-3 source datasets. So it tends to have faster convergence speed if the similar tasks are effectively identified at the initial stage, as demonstrated in Rover. The all-one strategy are easy to be disturbed by dissimilar data but can fully utilize information, so it may have a higher convergence value at later stage. The linear-change strategy is relatively more stable because it takes the advantages of the above two strategies into account.

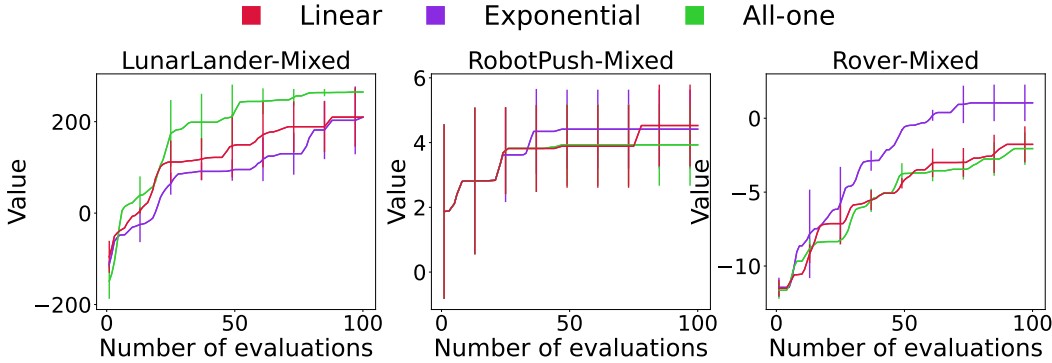

Figure 9: Sensitivity Analysis of Weight Change Strategy

**Decay factor** $\gamma$. The decay factor $\gamma$ is used to control the influence of source tasks. A higher decay factor $\gamma$ results in a quicker decay of source tasks' influence, thus making the node potential rely more on the target task's data. However, an excessively rapid decay of source task lead unstable evaluation outcomes and under-utilization of source task data. Specially, for the nodes preferred by source tasks, if they can't be selected as the sampling node at first, their potential will decrease rapidly and these nodes will be less likely to be selected. The analysis experiment of $\gamma$ can be seen in figure 10, and the result is as expected. The curves of MCTS-transfer with $\gamma = 0.99$ and $\gamma = 1.0$ overlap, better than $\gamma = 0.1$ in most cases, due to data exploitation. And an appropriate decay factor will help to combine information from source and target tasks to accelerate optimization.

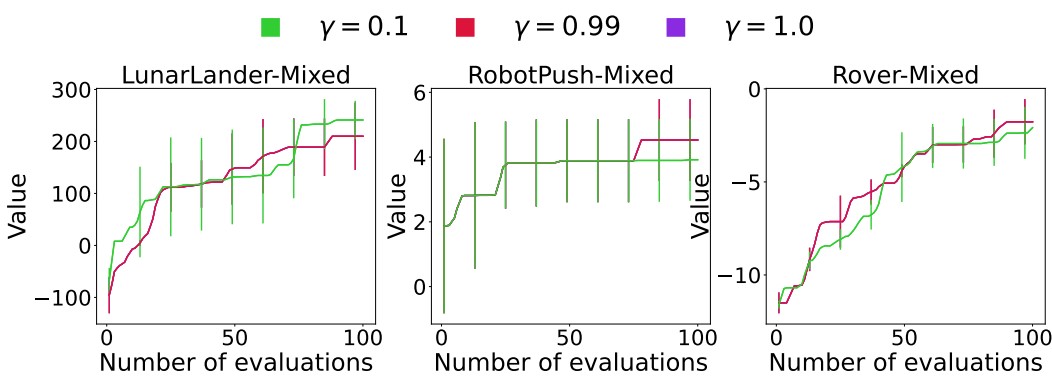

Figure 10: Sensitivity Analysis of Decay Factor $\gamma$

**Important source task ratio** $\alpha$   In linear-change strategy, we can set $\alpha$ to determine the ratio of important source tasks that have high weights. We choose $\alpha = 10\%, 50\%$ and $100\%$ and he result is shown in Figure 11. A higher $\alpha$ means that more source tasks are influencial on the evaluation of the tree node potential. However, this also increases the risk of the interference from dissimilar data. MCTS-transfer with a smaller $\alpha$ only trusts the most similar tasks and under-utilizing information from the source task, leading to slower convergence in early stage.

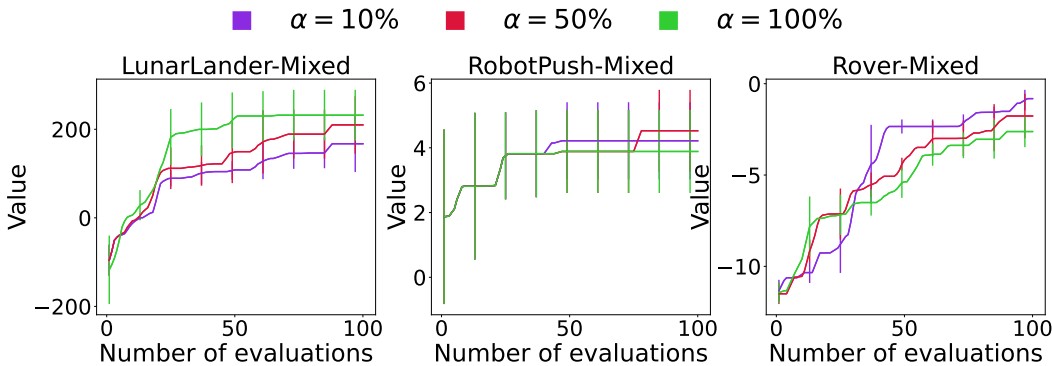

Figure 11: Sensitivity Analysis of Important Source Task Ratio $\alpha$

**Exploration factor** $C_p$    $C_p$ controls the degree of exploration. A large $C_p$ prompts the MCTS-transfer to select less promising regions, enhancing exploration. As shown in the Figure 12, too big $C_p$ results in the reduction of exploit useful information, as the algorithm overexploits less promising areas, leading to slower convergence.

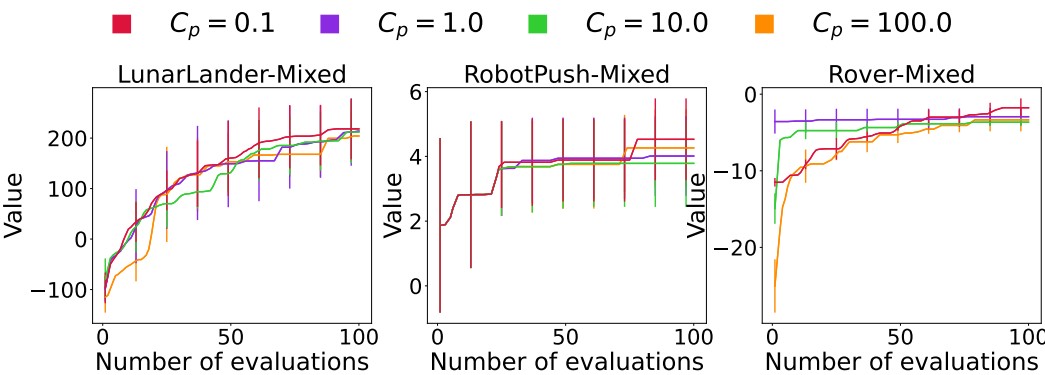

Figure 12: Sensitivity Analysis of Exploration Factor $C_p$

**The splitting threshold** $\theta$    The threshold $\theta$ controls the depth of the tree: a node is only allowed to be further divided when it contains more solutions than $\theta$. A smaller $\theta$ leads to a deeper tree. In our experiment (Figure 13), $\theta = 100$ means the tree nodes won't be splitted in the optimization, so the suggesting search space will be bigger, which is less efficient.

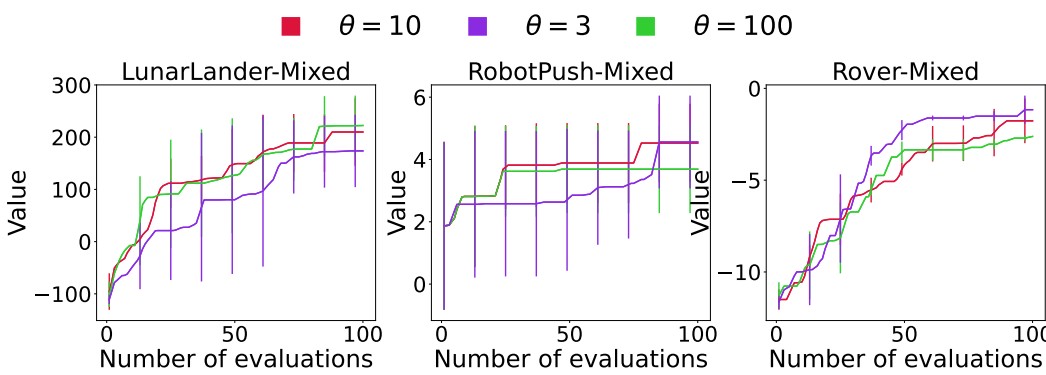

Figure 13: Sensitivity Analysis of The Splitting Threshold $\theta$

**Binary classifier** The binary classifier decides the boundary of "good" and "bad" clusters. We try the following classifiers: Logistic Regression and SVM (with rbf, linear, or poly kernel). According to the results in Figure 14, SVM with linear kernel and Logistic Regression give more effective search space partition. We can choose a binary classifier with higher partition efficiency according to specific scenarios.

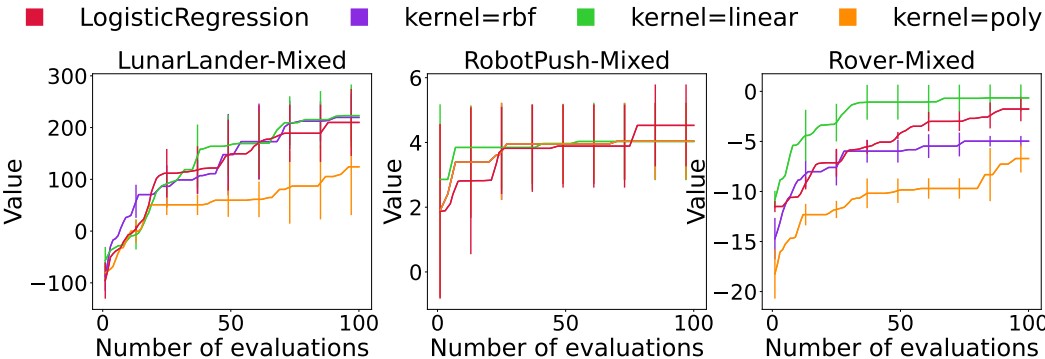

Figure 14: Sensitivity Analysis of Binary Classifier

# E    Detailed experimental results

The evaluation result curves on BBOB, real-world problems and Design-Bench are summarized in figures 15 and 16. The curves of HPOB in similar setting and mixed setting are shown in figure 18 and figure 19, and the format of subfigure titles is {search space id}-{dataset id}-{Simialr/Mixed}.

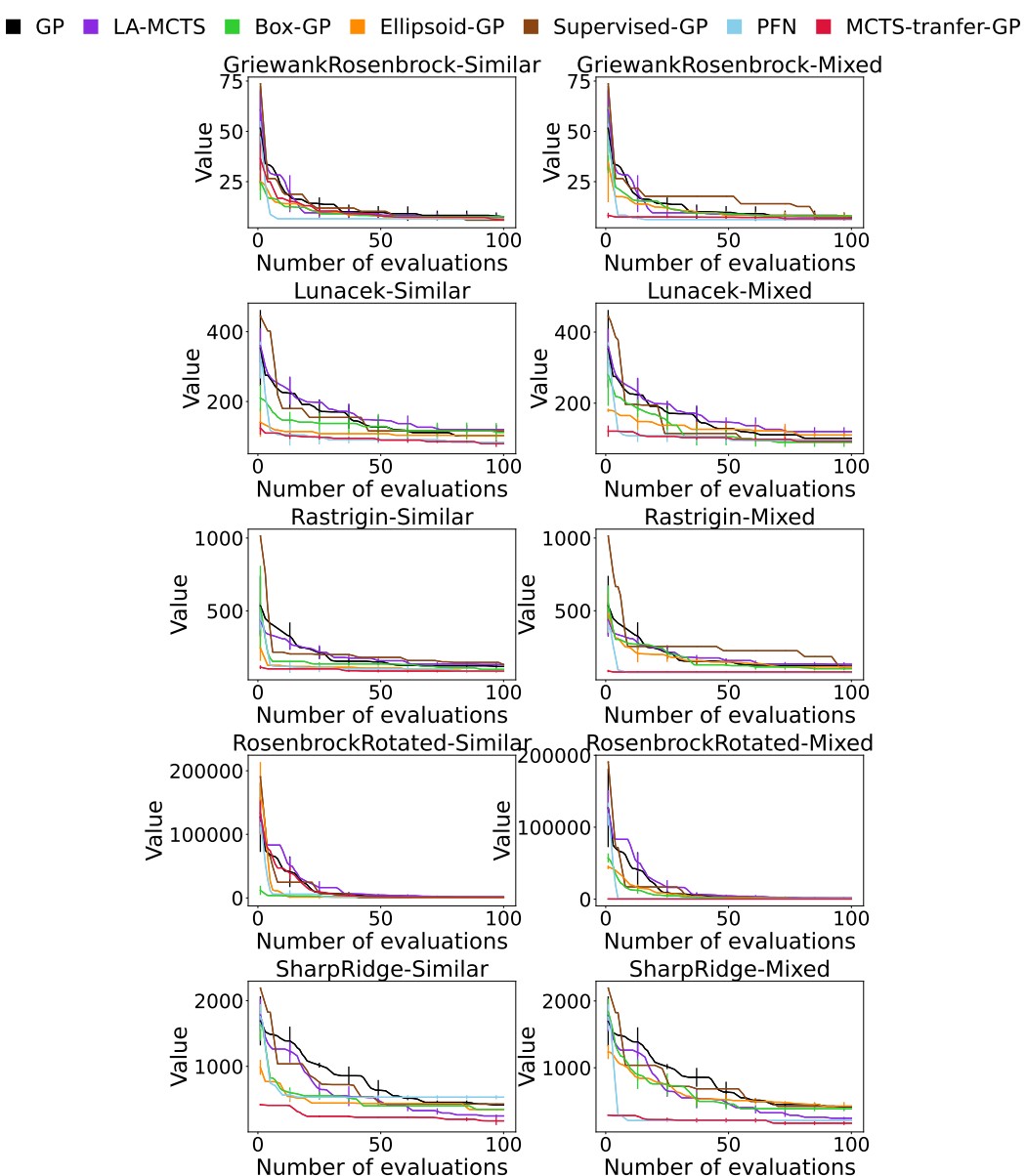

Figure 15: Evaluations of MCTS-transfer and other algorithms on BBOB

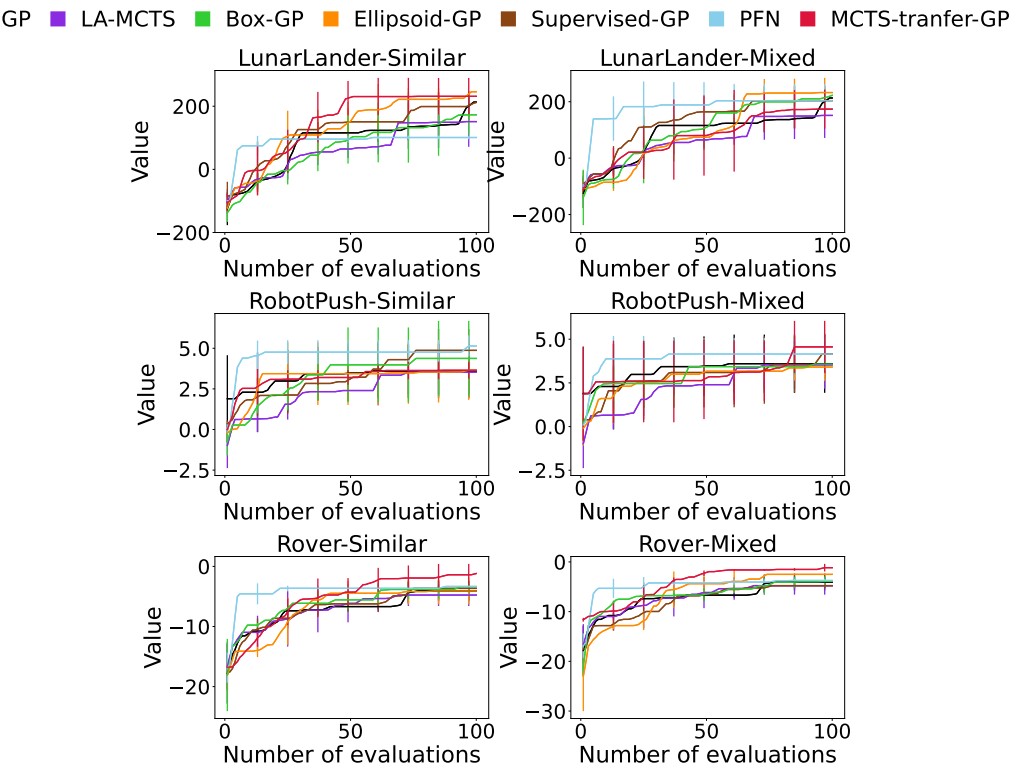

Figure 16: Evaluations of MCTS-transfer and other algorithms on Real-world Problems

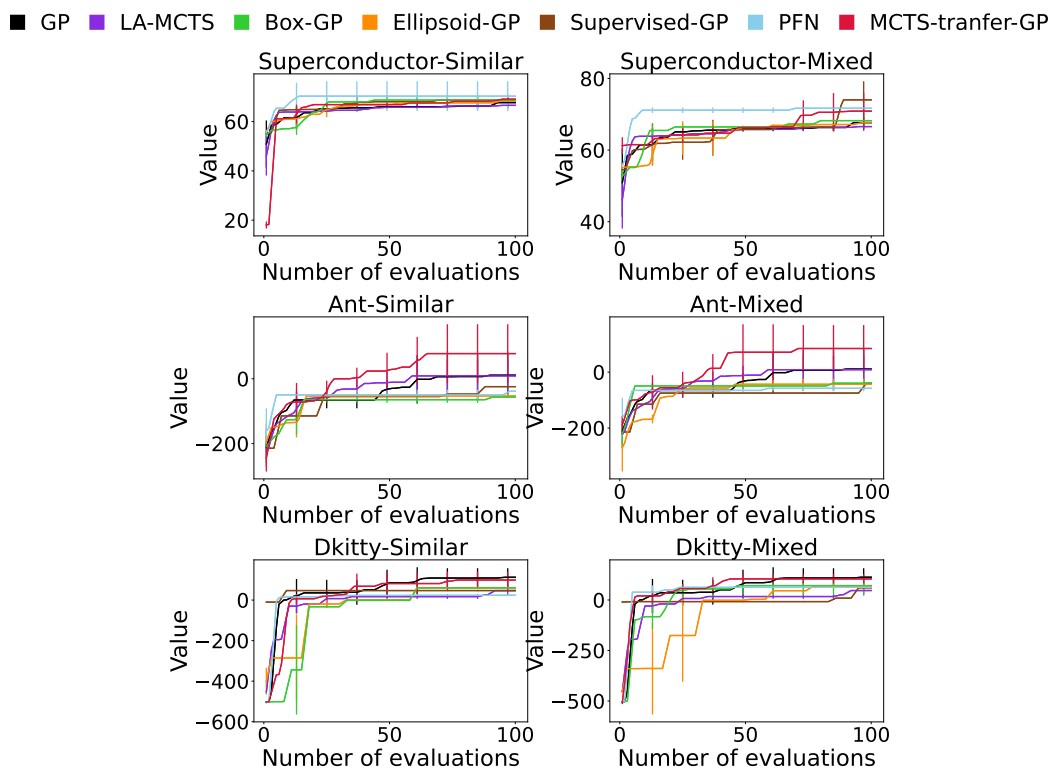

Figure 17: Evaluations of MCTS-transfer and other algorithms on Design-Bench

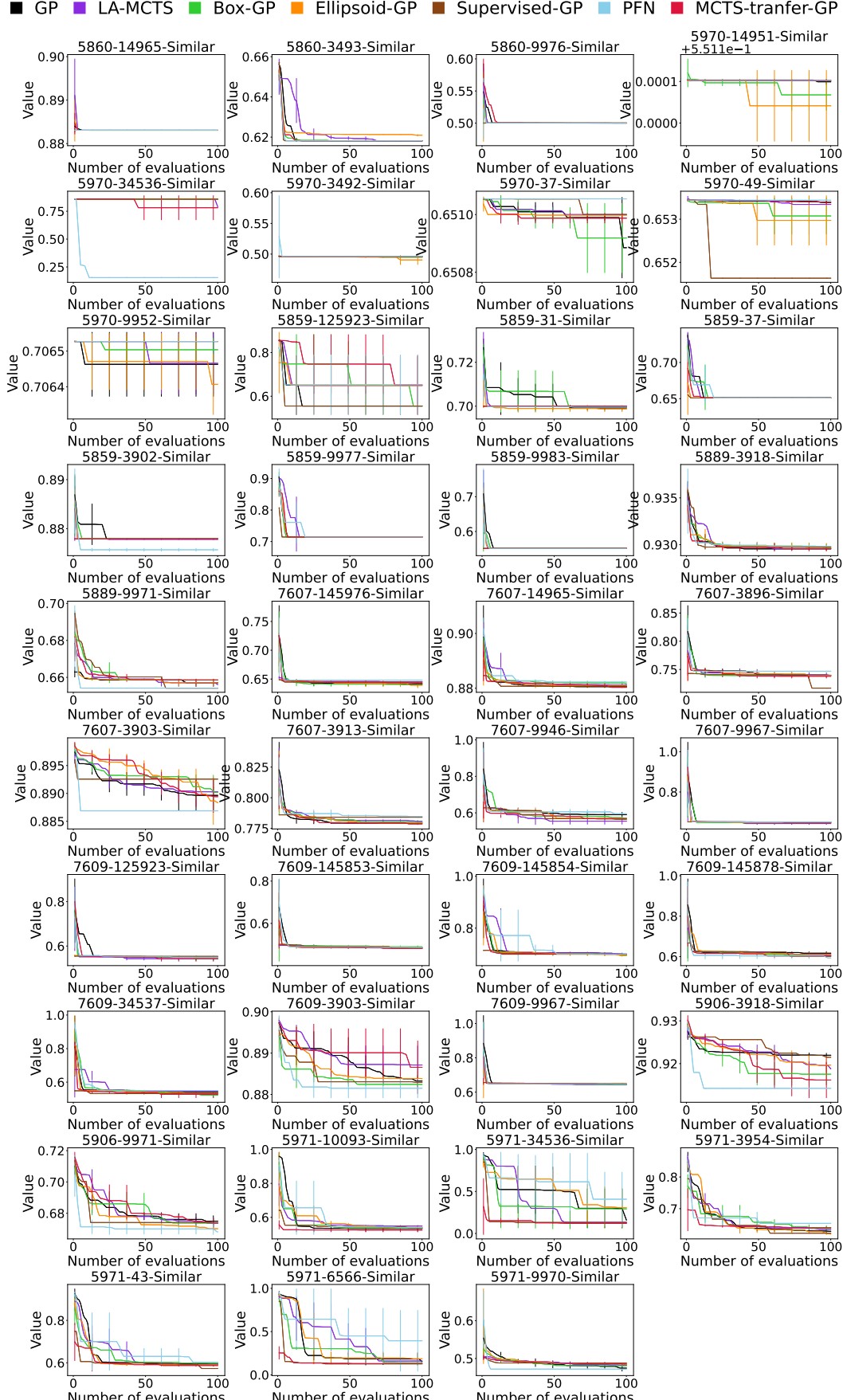

Figure 18: Evaluations of MCTS-transfer and other algorithms on HPOB in similar setting

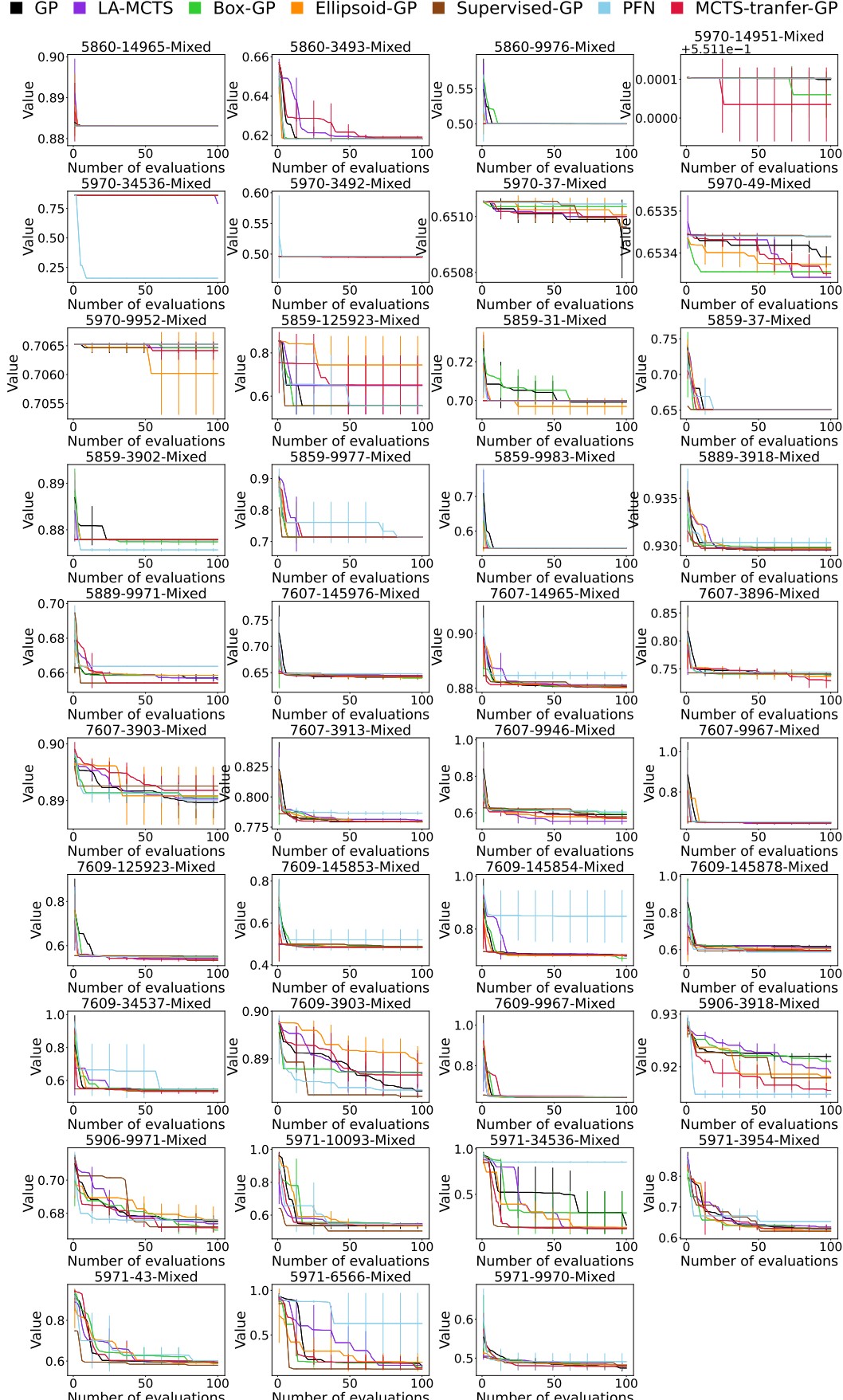

Figure 19: Evaluations of MCTS-transfer and other algorithms on HPOB in mixed setting

# F   Details of runtime analysis

In order to analysis the additional computational overhead introduced by MCTS-transfer, we calculate the time cost and the corresponding proportion of each component, and record the frequency of subtree reconstruction in each iteration.

Specifically, we divide MCTS-transfer into three main components: evaluation, backpropagation and reconstruction. The evaluation component includes the time required for surrogate model fitting, candidate solution selection and evaluation, which is a common component shared by all compared optimization algorithms. Backpropagation and reconstruction components are the two principal modules specific to MCTS-transfer.

We test MCTS-transfer on BBOB benchmark and three real-world tasks (i.e., LunarLander, Robot-Push, and Rover). Among them, the evaluation process of the real-world tasks is more time-consuming and the evaluation cost of BBOB are relatively cheaper.

As illustrated in Figure 20, the additional computational burden introduced by MCTS-transfer (i.e., backpropagation and reconstruction) represents a relatively minor fraction of the total runtime, particularly in the three real-world scenarios. These scenarios precisely exemplify the computationally intensive cases that transfer BO is designed to address, wherein MCTS-transfer demonstrates small additional computational overhead.Furthermore, our result reveals that the average frequency of tree reconstructions is low, with the corresponding reconstruction time being almost negligible when compared to the evaluation time.

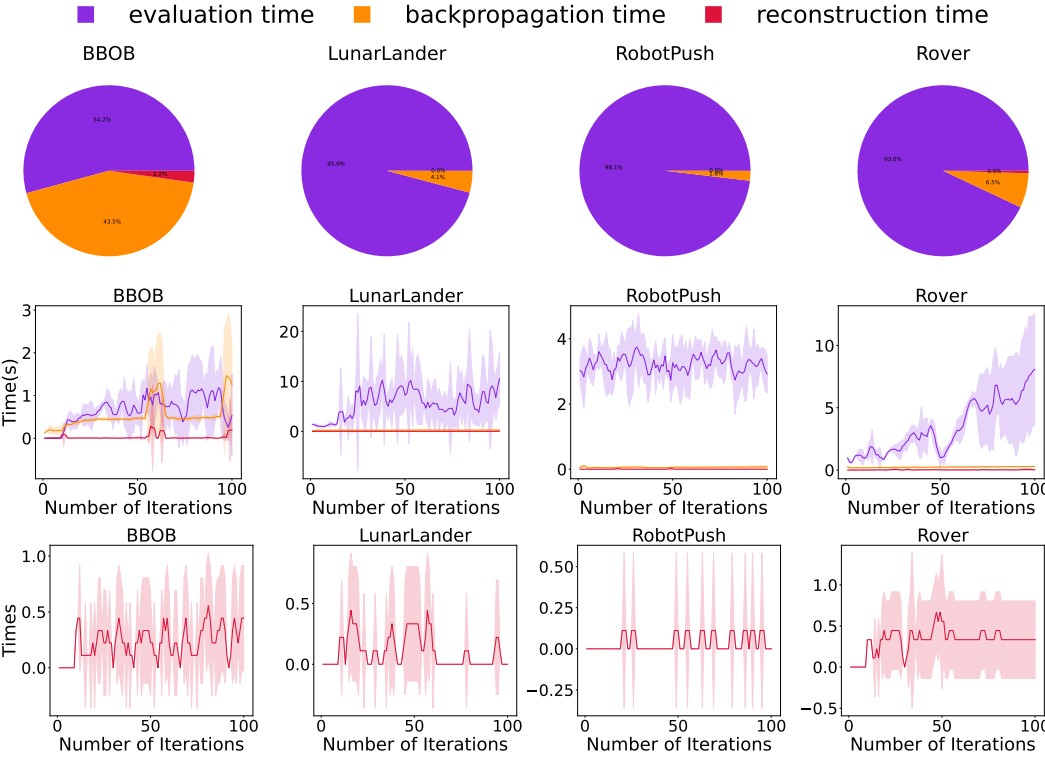

Figure 20: Time cost of different components in MCTS-transfer. The top row: the time proportion of components in optimization. The middle row: the time cost of components in each iteration (avg±std). The bottom row: the number of subtree reconstructions/reconstruction times in each iteration (avg±std).

# G Visualization of Weight Change Curve of Source Tasks

Here we show the weight change curve of three mixed real-world problems. We set the weight assignment stategy to be the linear change strategy with $\alpha = 0.5$ and the decay factor to be $\gamma = 0.99$. Figure 21 shows that the weights of real-world problem and similar sphere problem exceed those of dissimilar sphere problem in most cases, regardless of any inconsistencies in initialization. The results prove that the weight change strategy can prioritize similar source task data, which will lead to more accurate node potential evaluation and search space partition.

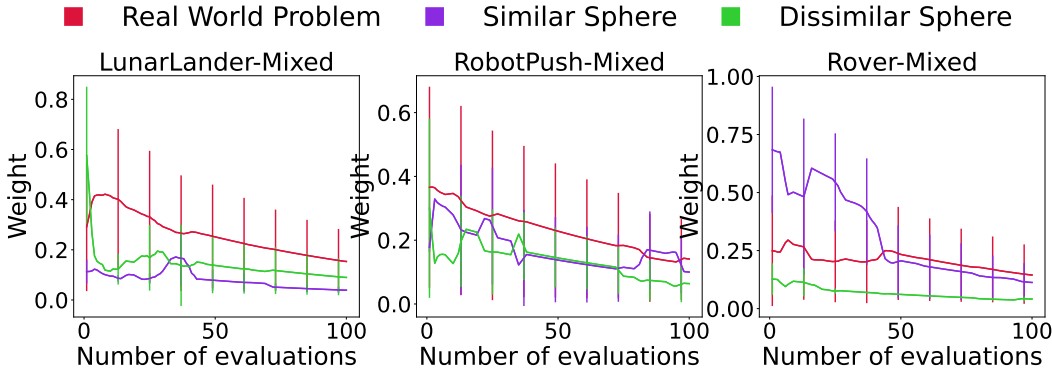

Figure 21: Weight Changes: MCTS-transfer

