# OpenReview forum: "Monte Carlo Tree Search based Space Transfer for Black Box Optimization"
_NeurIPS.cc/2024/Conference — NeurIPS 2024 spotlight_

### Official Review · Reviewer_Tjsz · 2024-06-20

**Soundness:** 3
**Presentation:** 3
**Contribution:** 3
**Rating:** 6
**Confidence:** 4

**Summary:**

This paper propose a search space transfer learning method based on Monte Carlo tree search, called MCTS-transfer, to iteratively divide, select, and optimize in a learned subspace. It can provide a well-performing search space for warm start for the target problem based on the source problems. It adaptively identify and leverage the information of similar source tasks to reconstruct the search space during the optimization process. Experiments on many situations have demonstrated the effectiveness of the algorithm.

**Strengths:**

1. The paper is well written, and is easily to understand.

2. The experiments are conducted extensively and demonstrate the effectiveness of the algorithm across numerous application scenarios.

3.  Introducing Monte-Carlo tree search into the search space transfer problem is quite novel, and this paper modifies some operations of MCTS based on its own application scenarios to make it work better.

**Weaknesses:**

1. It would be beneficial to analyze the algorithm's running time and computational cost if possible.

2. In MCTS of AlphaZero, state value $v$ is used to predict the expected future reward from the current state to the end. In this paper, the evaluation of current state $p_m$, which can be treated as a summary of historical iterative path from the root to the current, is used as $v$. Can you discuss the differences between the two? What are the potential impacts to use historical $p_m$ to represent the future expected $v$?

**Questions:**

Please refer to the weakness part.

**Limitations:**

Please refer to the weakness part.

---

> ### Author Rebuttal · Authors · 2024-08-07
>
> Thanks for your valuable and constructive comments. Below please find our responses.
>
> ### Q1: Running time and computational cost analysis
> Thanks for your valuable suggestions. Please refer to Q1 in general response.
>
> ### Q2: Discussion between the state value in MCTS of AlphaZero and MCTS-Transfer
>
> Good points! Thank you very much for your interesting and insightful questions.  AlphaZero [1] is designed to master complex games through self-play without relying on human knowledge or guidance. MCTS plays a crucial role in AlphaZero's decision-making process, whose state value is used to predict the expected future reward from the current state to the end, rather than the historical information in our paper. Different from that a state's future reward of AlphaZero can be obtained through multi-step simulations, i.e., alternating decisions through self-play, evaluation values in BBO can only be obtained through actual evaluations. Consequently, we utilize historical information in our paper. We have observed that some recent look-ahead BO works [2-3] have been used to predict the expected value of future steps in BBO problems, which have the potential to be applied in MCTS-Transfer as estimates for state values to further improve the performance. We will incorporate this discussion into our revised paper. Thank you for bringing this to our attention.
>
> [1] A general reinforcement learning algorithm that masters chess, shogi and Go through self-play. Science, 2018.
>
> [2] Practical Two-Step Look-Ahead Bayesian Optimization. NeurIPS, 2019.
>
> [3] Accelerating Look-ahead in Bayesian Optimization: Multilevel Monte Carlo is All you Need. ICML, 2024.

---

### Official Review · Reviewer_jvcN · 2024-07-10

**Soundness:** 3
**Presentation:** 3
**Contribution:** 3
**Rating:** 7
**Confidence:** 4

**Summary:**

This paper proposes a search space transfer learning method based on Monte Carlo tree search (MCTS) called MCTS-transfer, which aims to accelerate the optimization process in computationally expensive black-box optimization problems.

**Strengths:**

- Originality: The integration of MCTS with search space transfer learning is a novel approach, which addresses the need for improved convergence in black-box optimization problems.
- Clarity: The paper is generally well-written and structured, making the methodology and results comprehensible.

**Weaknesses:**

- Technical Depth: The paper lacks sufficient technical depth in explaining the underlying mechanics of MCTS-transfer. For instance, the specifics of how MCTS iteratively divides and selects subspaces are not clearly detailed.
- Experimental Rigor: The experimental validation, although covering various scenarios, does not delve deeply into comparative baselines. The choice of baselines is limited and more recent advancements in the field should be included.
- Adaptability and Scalability: There is insufficient discussion on the adaptability of MCTS-transfer to different problem domains and its scalability to very large search spaces. The potential computational overhead and limitations in such scenarios are not adequately addressed.
- Theoretical Analysis: The theoretical analysis supporting the method's efficacy is minimal. It would be better if more rigorous proofs or detailed theoretical justifications would strengthen the paper.
- Reproducibility: While the results are promising, more details should be presented to ensure reproducibility. Key implementation details, parameter settings, and the codebase are missing, which are crucial for the validation of the proposed method.

**Questions:**

1. How does MCTS-transfer handle the computational overhead introduced by the MCTS component, especially in large and complex search spaces?
2. Can the authors provide more specific examples or case studies where MCTS-transfer significantly outperforms other methods?
3. What measures have been taken to ensure the reproducibility of the results? Is the code and dataset publicly available?

**Limitations:**

The authors should address the potential computational overhead and provide a more comprehensive discussion on the limitations of their method. Additionally, the impact of the method on different types of black-box optimization problems should be more thoroughly explored.

---

> ### Author Rebuttal · Authors · 2024-08-07
>
> Thanks for your valuable and constructive comments. Below please find our responses.
>
> ### Q1: How does MCTS iteratively divides and selects subspaces?
>
> We're sorry that we didn't make this part clear.
>
> - How does MCTS iteratively divide subspaces? As described in Section 3.1, the division of space corresponds to the expansion of nodes. Starting from the ROOT node, if the node is splittable, we cluster the samples into two clusters and divide them apart, leading to the birth of two child nodes, where the node with better cluster is the left child node. Then, following the sequence of depth-first-search, we repeatedly try to expand the node and visit the child nodes if it's splittable, iteratively dividing the space.
>
> - How does MCTS iteratively select subspaces? We select the subspace by the guidance of UCB. Starting from the ROOT node, if the node has two child nodes, we will choose the node with higher UCB as the next node to visit. Following the sequence, we will finally locate the target leaf node and find the subspace.
>
> ### Q2: Comparison with more advanced baselines
> Thanks for your comments. However, we believe that our compared baselines are comprehensive. We include non-transfer BO (basic GP, LA-MCTS), existing search space transfer methods (Box-GP, Ellipsoid-GP, Supervised-GP), and state-of-the-art algorithm (PFN). If you have some specific suggestions of additional baselines, we are happy to discuss it.
>
> ### Q3: Discuss the adaptability of MCTS-transfer to different problem domains and its scalability to very large search spaces
>
> Thanks for your valuable comments.
>
> - Adaptability to different problem domains. We add three new complex real-world problems from Design-Bench to show the adaptability of MCTS-transfer to different problem domains.
>
> - Scalability to very large search spaces. MCTS-transfer is suitable for solving large search spaces, because the subspaces with high potential can be gradually discovered and extracted, thus improving the optimization efficiency. The three new real-world problems from Design-Bench are relatively high-dimensional, among which Superconductor has 86 dimensions, Ant morphology has 60 dimensions, and D’Kitty Morphology has 56 dimensions. We believe the results can reflect the performance of MCTS-transfer in large search spaces.
>
> Please see Q2 in general response for details.
>
>
> ### Q4: Lack of theoretical analysis
>
> Thank you very much for your suggestions. We fully agree that theoretical analysis of MCTS-transfer is a very interesting topic. As also suggested by Reviewer x4CE, we provide two potential perspectives for theoretical analysis:
>
> - One point of analysis could be the transfer efficiency in transfer learning. A possible approach is to analyze the space size covered by MCTS partitions at the optimal points [1]. The expected conclusion is that, compared to constructing LaMCTS from scratch, MCTS partitions cover optimal points more efficiently with the same number of partitions.
>
> - Another point that can be theoretically analyzed is the regret bound. A possible approach is to use the error bounds of Gaussian process regression [2] and the characteristics of MCTS space partitions to bound the instantaneous regret at each step.
>
> We will add this discussion into our main paper and leave it as our future work. Thank you.
>
> [1] Multi-Objective Optimization by Learning Space Partitions. ICLR, 2022.
>
> [2] Gaussian Process Optimization in the Bandit Setting: No Regret and Experimental Design. ICML, 2010.
>
> ### Q5: Reproducibility issue
>
> Our code has been provided in the supplemental materials. All the implementation details can be found in it. Besides, the parameter settings and data collection methods are provided in appendix A.1 and A.3, respectively.
>
> ### Q6: How does MCTS-transfer handle the computational overhead introduced by the MCTS component, especially in large and complex search spaces?
>
> We acknowledge that building and maintaining MCTS in high-dimensional and complex search spaces will bring additional computational overhead. However, as shown in Figure 1 in the PDF file, the time of tree backpropagation and reconstruction seems minor compared to evaluation time. Additionally, if one wants to further reduce the time cost on MCTS, we can set the leaf size $\theta$ higher to reduce the tree depth, or choose a fast binary classifier such as LinearRegression to divide the space.
>
> ### Q7: Provide more specific examples or case studies where MCTS-transfer significantly outperforms other methods
>
> MCTS-transfer is suitable for expensive BBO, especially when we are uncertain about which tasks are similar to the target task. Our method can automatically identify the most relevant tasks and give more considerations on them when constructing the subspaces. Even if none of the source tasks are considered similar, MCTS-transfer can still correct the search direction by relying more on the collected target task data just as the motivating case in section 4.1 demonstrated. Hyperparameter optimization is a specific example in practice, where we don't know any properties in advance. However, we can collect other optimization trajectories on the same domain, and MCTS-transfer will automatically identify the relevant part to accelerate optimization.
>
> ### Q8: Provide a more comprehensive discussion on the limitations of their method
>
> Apart from lacking theory analysis and accurate task similarity measures, the limitations of our work include that it cannot handle search space transfer tasks for problems with different domains or with different dimensions. We will further improve the algorithm in this way by learning embedding space in future work. We will add this part in the new version.

---

> > ### Comment · Reviewer_jvcN · 2024-08-08
> >
> > Thanks for your detailed response. My concerns have been addressed. I have raised the score and confidence accordingly.

---

> > > ### Author Response · Authors · 2024-08-08
> > >
> > > Thanks for your feedback! We are glad to hear that your concerns have been addressed. We will make sure to include the added results and discussion in the final version. Thank you.

---

### Official Review · Reviewer_ygQe · 2024-07-12

**Soundness:** 4
**Presentation:** 3
**Contribution:** 4
**Rating:** 7
**Confidence:** 4

**Summary:**

This paper proposes a new space transfer method for black-box optimization by using MCTS. The search space is divided by MCTS, and the data from source tasks are used to help evaluate the value of each node of the tree. The similarity between the source and target tasks is considered and adjusted dynamically. The authors performed experiments on various problems, and compared the propose method with state-of-the-art methods.

**Strengths:**

The idea of the proposed method so-called MCTS-transfer is interesting and also natural. It extends the LA-MCTS method by using transfer learning: The data points of the source tasks are used to help evaluate the value of a node in the tree with weights related to their similarity to the target task. The similarity is updated using new sampled points from the target task and will become more accurate gradually.

In the initialization stage, the data of source tasks are used to construct a tree by clustering and binary classification. Particularly, the data points in a node are clustered into two clusters, where the cluster with better average objective value is treaded as “good” and the other cluster is “bad”. A binary classifier is then used to divide the space represented by the node into two parts. In the optimization stage, the proposed method uses MCTS to select one leaf node, and optimizes within the search space the leaf represents. The newly sampled points are further used to update the leaf node, e.g., expand the leaf.

The most interesting thing is the calculation of the potential value of each node, which is a weighted sum of objective values of sampled points from both source and target tasks. The weights consider the similarity between the source and target tasks; more similar, larger the weights. The authors calculated the weights by using the distance between the best points of the tasks, which will be updated after new data points of the target task are sampled. This makes the weights (or the similarity between the source and target tasks) adjustable during the optimization process.

The experiments are extensive. The authors compared state-of-the-art methods of transferring the search space, and some other recent related methods. The problems considered include BBOB, hyper-parameter optimization, and real-world problems. The results generally show the superior performance of MCTS-transfer. The authors also did various sensitivity analyses. The paper is overall well written, and easy to follow. The codes are provided.

Overall, I think this work can provide a good complement to the space transfer method for black-box optimization.

**Weaknesses:**

The authors considered the number of evaluations in the experiments. This is OK. But I also want to see the running time comparison of each iteration, which will be useful in practice. It seems that the proposed method will cost more time, as it will use the procedure of Treeify to check the feasibility of the tree, i.e., whether a right child node has a larger value than the left one.

In the right subfigure of Figure 1(b), no blue line?

In the right subfigure of Figure 2(b), the PFN method is better than the proposed MCTS-transfer. I’d like to see some discussion. As MCTS-transfer can be equipped with any BO algorithm, can it achieve better results by combining advanced BO algorithms?

Lines 346-347, I cannot understand “we can still see the obvious strength in exploring the optimal solution at later stage,” can you give some explanation?

I suggest moving the pseudo-code of Algorithm 1 (Treeify) to appendix. Instead, you can include more experimental results in the main paper.

For the hyper-parameters \gamma and \alpha in equation 3 and 4, how to set them in practice?

Though the paper is overall well written, there are still some types.

-- line 310: 3 search space transfer algorithms -> three search space transfer algorithms

-- line 311: figure 1 -> Figure 1; line 322: figure 1 -> Figure 1; Please check throughout the paper.

-- line 330: The sentence “The detailed experimental results …” is redundant.

-- line 334: mcts-transfer -> MCTS-transfer

-- line 336: “equal” -> “reach”

-- line 337: “surprising performance”, I suggest using “superior performance”. The results are good, but not surprising.

-- line 339: “We test” -> “we test”

-- line 344: “in the RobotPush” -> “in RobotPush”

-- line 354: “close final and random final”?

-- line 360-361: in D -> in Appendix D; but it can -> it can

-- line 600: missing the period

-- line 621: \alpha -> and \alpha

-- line 643-645: the sentence is not well written.

-- line 641, 646: check the missing and redundant blank

-- Caption of Figure 13: Real-World Problem -> Real-World Problems

**Questions:**

See weakness.

**Limitations:**

Yes.

---

> ### Author Rebuttal · Authors · 2024-08-07
>
> Thank you for carefully reviewing our paper and providing constructive comments, which have helped improve the work a lot. We are very glad that you appreciate our work. Below please find our responses.
>
> ### Q1: The running time comparison of each iteration
>
> Thank you for your valuable suggestions. Please refer to Q1 in general response.
>
> ### Q2: Explain the lack of blue line in the right subfigure of Figure 1(b)
>
> In Sphere2D, we consider mixed settings and dissimilar settings. As demonstrated in line 307-309, we use $D\_{(-5,-5)},D\_{(5,-5)},D\_{(5,5)}$ as source datasets in mixed setting, and remove the most similar task $D\_{(5,5)}$ in dissimilar setting. Figure 1(b) shows the curve of source task weight changes during the optimization process. As there are only two source tasks in dissimilar setting, only two lines are shown in right subfigure of Figure 1(b).
>
> ### Q3: In Figure 2(b), PFN is better than MCTS-transfer; Combining MCTS-transfer with advanced BO algorithms to achieve better performance.
>
> Thanks for your comments, but there are some misunderstandings that need to be clarified. In Figure 2(b), PFN just converges faster in the early stages, but performs worse than our MCTS-transfer after about 70 iterations. Our algorithm finally achieved better results as well.
>
> We appreciate your suggestion that combining MCTS-transfer with other advanced BO algorithms, which is an interesting idea and can further strengthen our work. According to your suggestion, we combine MCTS-transfer with PFN and compare it with other algorithms in Figure 2(b). The results can be found in the right subfigure of Figure 2 in the PDF file. We find that MCTS-transfer-PFN makes further improvements compared to MCTS-transfer-GP and PFN, which shows the versatility of MCTS-transfer. We will include this experiment in our revised paper. Thank you very much.
>
> ### Q4: Explain line 346,347
>
> We are sorry for not expressing it clearly.
> In RobotPush, MCTS-transfer has no advantage in the initial stage, but it can still effectively divide the search space and speed up optimization, as clearly shown in similar settings. In the later stage of optimization in similar setting, the speed of MCTS-transfer for finding the optimal solution exceeds all baselines, which may come from the efficient utilization of source task data by reasonable node potential evaluation, node expansion, and tree reconstruction.
>
> ### Q5: How to set $\gamma$ and $\alpha$ in practice?
>
> $\gamma$ is a decay factor controlling the weight decay of source tasks, and $\alpha$ is to determine the source task ratio with high weights. We set $\gamma=0.99$ and $\alpha=0.5$ by default. Generally, if the source tasks are very relevant to the target task, i.e., very similar or important, you can set $\gamma$ and $\alpha$ higher; if the source tasks are diverse, $\alpha$ should be set lower to prevent disturbance from unimportant tasks.
>
> ### Q6: Remove treeify to appendix; Some typos in paper
>
> Thank you very much for carefully pointing out the typos and providing your valuable suggestions on the paper organization. We will revise the paper carefully according to your suggestions.

---

> > ### Comment · Reviewer_ygQe · 2024-08-12
> >
> > The responses have answered my questions and further confirmed my rating.  Thank you.

---

### Official Review · Reviewer_x4CE · 2024-07-12

**Soundness:** 3
**Presentation:** 4
**Contribution:** 3
**Rating:** 7
**Confidence:** 3

**Summary:**

The paper proposes a tree-based search space division to enable transfer across different instances of related optimisation problems.
The authors propose both a scheme to divide the search space in a hierarchical fashion based on training task samples as well as a way of weighing the resulting subspaces against each other to acquire new evaluations for the task at hand. Since the similarity between the training tasks and the current task is updated as samples are acquired, the tree rankings have to be continuously updated as well.
The derived algorithm is evaluated on both an illustrating toy example and several more challenging benchmarks. The appendix contains further ablations regarding design choices and hyperparameter selections.

**Strengths:**

I found the paper very well-presented and easy to follow. The main ideas were clearly laid out and the overall structure made sense to me.

To my knowledge, the proposed approach is novel. Given that Bayesian Optimization is usually applied in tasks with limited evaluation budgets, search space transfer seems like a promising avenue of allocating these limited budgets more efficiently. It also circumvents the scaling issues that approaches based on synthetic data points have.

The empirical evaluation is sound and there are detailed ablation studies supporting the authors claims.

**Weaknesses:**

Arguable the biggest weakness of the paper is the lack of theoretical analysis of the provided algorithm. However, given the depth of the empirical evaluation, I believe this can be left as future work.

I also feel that a comment on the runtime of the proposed framework would be helpful. MCTS-transfer requires the (re-)construction and update of an entire search tree as well as the training of several classifiers.

Beyond this, some minor points for consideration are:
- From the description it is unclear, when the search space clustering (i.e. the subspace classifier) is updated. Does this only happen during tree reconstruction or during back-propagation as well?
- I would not name LunarLander, RobotPush, and Rover as real-world problems. While the search spaces are higher dimensional, the problems themselves are relatively simple

**Questions:**

1. How often is reconstruction of the search tree required? Would it be possible to add an analysis of how frequently the tree has to be rebuild as this presumably is a large factor in the runtime of the algorithm itself?

2. Could the authors comment on how the search-space division is done for conditional search spaces (e.g. in the hyperparameter optimization settings)?

**Limitations:**

The authors point towards future work and more accurate task similarity measures as future work but beyond this do not discuss the limitations of their work.

---

> ### Author Rebuttal · Authors · 2024-08-07
>
> Thank you for carefully reviewing our paper and providing constructive comments, which have helped improve the work a lot. Below please find our responses.
>
> ### Q1: Lack of theory
>
> Thank you very much for your suggestions. We fully agree that theoretical analysis of MCTS-transfer is a very interesting topic. Here, we provide two potential perspectives for theoretical analysis:
>
> - One point of analysis could be the transfer efficiency in transfer learning. A possible approach is to analyze the space size covered by MCTS partitions at the optimal points [1]. The expected conclusion is that, compared to constructing LaMCTS from scratch, MCTS partitions cover optimal points more efficiently with the same number of partitions.
>
> - Another point that can be theoretically analyzed is the regret bound. A possible approach is to use the error bounds of Gaussian process regression [2] and the characteristics of MCTS space partitions to bound the instantaneous regret at each step.
>
> We will add this discussion into our main paper and leave it as our future work. Thank you.
>
> [1] Multi-Objective Optimization by Learning Space Partitions. ICLR, 2022.
>
> [2] Gaussian Process Optimization in the Bandit Setting: No Regret and Experimental Design. ICML, 2010.
>
> ### Q2: Lack of runtime analysis and reconstruction frequency analysis
>
> Thank you for your valuable suggestions. Please refer to Q1 in general response.
>
> ### Q3: When is the search space clustering updated?
>
> We are sorry for not making it clear. The update of tree clustering happens both in backpropagation and reconstruction stage. In backpropagation stage, after we get a new sample $(\boldsymbol{x}\_t, f(\boldsymbol{x}\_t))$ in node $m$, we will update the node status along the path from $m$ to ROOT node, including the data contained in the node, the number of visits, and whether it is splittable. To check whether the node $m$ is splittable, we need to update the clustering of the search space $\Omega\_m$. If $m$ contains more than $\theta$ samples and the samples can be clustered apart, we will expand $m$ into 2 child nodes. In reconstruction stage, we will prune the unqualified subtrees and reconstruct them.  During the subtree reconstruction, the clustering of the search space will also be updated. We will revise to make it clear. Thank you.
>
> ### Q4: The real-world problems are relatively simple
>
> Thanks for pointing this out. We will rename these problems as non-synthetic tasks. To further demonstrate the performance of MCTS-transfer in complex real-world problems, we add three new problems from Design-Bench [1]. The results still show the superiority of MCTS-transfer. Please refer to Q2 in general response.
>
> [1] Design-Bench: Benchmarks for Data-Driven Offline Model-Based Optimization. ICML, 2022
>
> ### Q5: How to do the search-space division for conditional search spaces?
>
> Thanks for your question. To apply our method to the conditional search space, we may apply the following process. For conditional optimization, we consider the problem $\min \_{\boldsymbol{x} \in \mathcal{X} \subset \mathbb{R}^{d}} f(\boldsymbol{x})$. Specifically, the search space is tree-structured, formulated as $\mathcal{T}=\\{V,E\\}$, where $v\in V$ is a node representing subspace and $e\in E$ is an edge representing condition. The objective function is also defined based on $\mathcal{T}$, formulated as $f\_{\mathcal{T}}(\boldsymbol{x}):=f\_{p\_{j},\mathcal{T}}(\boldsymbol{x}|{l\_{j}})$, where $p\_j$ is a condition and $\boldsymbol{x}|l\_j$
> is the restriction of $\boldsymbol{x}$ to $l\_j$ [1]. In the pre-learning stage, it builds subtrees for each $v\in V$ and generates the MCTS model $\mathcal{T'}$ based on $\mathcal{T}$. In each iteration, followed by UCB value, it finds the target node $m$ located in the subtree of $v$ with condition $p\_i$,  optimizes in $\Omega\_m$, selects and evaluates the candidate using $f\_{p\_{i},\mathcal{T}}(\boldsymbol{x}|{l\_{i}})$. After that, it updates the task weights and node potential in the whole tree $\mathcal{T'}$ and tries to reconstruct the tree. Note that the tree reconstruction only happens in each subtree of $v\in V$. We will revise to add some discussion.
>
> [1] Additive Tree-Structured Covariance Function for Conditional Parameter Spaces in Bayesian Optimization. AISTATS, 2020.
>
>
> ### Q6: The limitations of the work
>
> Thanks for your suggestions. Apart from lacking theoretical analysis and accurate task similarity measures, the limitations of our work include that it cannot handle search space transfer tasks for problems with different domains or with different dimensions. We will further improve the algorithm in this way by learning embedding space in future work. We will add this to our revised paper. Thank you.

---

> ### Comment · Reviewer_x4CE · 2024-08-12
>
> I thank the authors for their detailed response which have answered my questions and confirmed my rating.

---

### Author Rebuttal · Authors · 2024-08-07

We are very grateful to the reviewers for carefully reviewing our paper and providing constructive comments and suggestions. Our response to individual reviewers can be found in the personal replies, but we also would like to make a brief summary of revisions about writing, discussion, and experiments for your convenience.

Writing:

- We have revised some typos and improved some expressions.

Discussions:

- We discuss the theoretical analysis of MCTS-Transfer, according to the suggestions of Reviewers x4CE and jvcN.

- We discuss how to apply MCTS-Transfer to the conditional search space, according to the suggestions of Reviewer x4CE.

- We discuss the differences in MCTS between AlphaZero and MCTS-transfer, according to the suggestions of Reviewer Tjsz.

Experiments:

- We analyze the runtime of MCTS-transfer, according to the suggestions of Reviewers x4CE, ygQe, and Tjsz.

- We add three new real-world problems from Design-Bench to compare the performance of algorithms in complex and high-dimensional real-world problems, according to the suggestions of Reviewers x4CE and jvcN.

- We equip MCTS-transfer with PFN on mixed real-world problems, demonstrating the versatility of MCTS-transfer combining with other advanced BO algorithms, according to the suggestions of Reviewer ygQe.

For the important questions many reviewers concerned about, we make general responses here.

### Q1 Runtime analysis

We conduct a comprehensive analysis of the runtime proportion of each component of MCTS-transfer and quantify the subtree reconstruction times on BBOB benchmark and three non-synthetic tasks (i.e., LunarLander, RobotPush, and Rover).  We divide MCTS-VS into three main components: evaluation, backpropogation and reconstruction.

- The evaluation component includes the time required for surrogate model fitting, candidate solution selection and evaluation. This component is common to all the compared optimization algorithms.
- Backpropagation and reconstruction constitute the two principal modules specific to MCTS-transfer.

As illustrated in Figure 1 of the PDF file, the additional computational burden introduced by MCTS-Transfer (i.e., backpropagation and reconstruction) represents a relatively minor fraction of the total runtime, particularly in the three non-synthetic scenarios. These scenarios precisely exemplify the computationally intensive cases that transfer BO is designed to address, wherein MCTS-Transfer demonstrates small additional computational overhead.

Furthermore, our result reveals that the average frequency of tree reconstructions is low, with the corresponding reconstruction time being almost negligible when compared to the evaluation time. We will include this discussion in our revised paper. Thank you.

### Q2 More problem domains/complex real-world problems
To further verify the performance of MCTS-transfer in complex real-world problems, we add the following three problems from Design-Bench [1], which is a suite of diverse and realistic tasks derived from real-world optimization problems.

- Superconductor: critical temperature maximization for superconducting materials.  This task is taken from the domain of materials science, where the goal is to design the chemical formula for a superconducting material that has a high critical temperature. The search space is a continuous space with 86 dimensions.

- Ant morphology: robot morphology optimization. The goal is to optimize the morphological structure of Ant from OpenAI Gym to make this quadruped robot to run as fast as possible. The search space is a continuous space with 60 dimensions.

- D’Kitty Morphology: robot morphology optimization. The goal is to optimize the morphology of D’Kitty robot to navigate the robot to a fixed location. The search space is a continuous space with 56 dimensions.

The data collection methods are consistent with existing real-world tasks in our paper, and the parameters remain unchanged. The results shown in the left sub-figure of Figure 2 in the PDF validate the significant advantages of MCTS-transfer in these complex, high-dimensional, and realistic tasks.

[1] Design-Bench: Benchmarks for Data-Driven Offline Model-Based Optimization. ICML, 2022.

**We will include all these results into the revision of our paper, and will revise the paper carefully according to your comments and suggestions. We hope that our response has addressed your concerns, but if we missed anything please let us know.**

---

### Decision · Program_Chairs · 2024-09-25

**Decision:**

Accept (spotlight)

**Comment:**

This paper proposes MCTS-transfer, which utilizes MCTS with a search space transfer method in black-box optimization problems. The core idea is to use MCTS to iteratively divide the search space and optimize within a learned subspace by incorporating a weighted factor from source tasks into target tasks. The weights are calculated based on the similarity between the source and target tasks, and these weights are further updated as the target task is optimized. This makes the weight adjustment adaptive and addresses the issue that previous black-box optimization methods did not consider cases where the source and target tasks are not similar. MCTS then reconstructs the search tree based on the newly calculated node potentials. The authors also conducted detailed empirical experiments across various benchmarks to demonstrate that MCTS-transfer outperforms previous methods.

This paper is well-written and well-structured, with sufficient background, clear motivation and methodology, and strong experimental results supporting the proposed method. One of the major concerns raised by reviewers was the time cost of reconstruction in MCTS-transfer. During the rebuttal, the authors provided additional experiments to demonstrate that this cost is almost negligible. These additional experiments should be included in the final version. I also suggest the authors include a figure providing an overview of the procedure at the beginning of Section 3 (before Section 3.1) to enhance readability. For example, the figure could illustrate the tree-forming process, such as how the root node $m$ is expanded, and display the node potential values within the tree to give readers a better understanding.

To sum up, this paper is technically solid in the field of black-box optimization. I recommend acceptance with a spotlight.